

# Discrete two dimensional Fourier transform in polar coordinates part II: numerical computation and approximation of the continuous transform

Xueyang Yao[1] and Natalie Baddour[2]

[1] Department of Systems Design Engineering, University of Waterloo, Waterloo, ON, Canada
[2] Department of Mechanical Engineering, University of Ottawa, Ottawa, ON, Canada

## ABSTRACT

The theory of the continuous two-dimensional (2D) Fourier Transform in polar coordinates has been recently developed but no discrete counterpart exists to date. In the first part of this two-paper series, we proposed and evaluated the theory of the 2D Discrete Fourier Transform (DFT) in polar coordinates. The theory of the actual manipulated quantities was shown, including the standard set of shift, modulation, multiplication, and convolution rules. In this second part of the series, we address the computational aspects of the 2D DFT in polar coordinates. Specifically, we demonstrate how the decomposition of the 2D DFT as a DFT, Discrete Hankel Transform and inverse DFT sequence can be exploited for coding. We also demonstrate how the proposed 2D DFT can be used to approximate the continuous forward and inverse Fourier transform in polar coordinates in the same manner that the 1D DFT can be used to approximate its continuous counterpart.

# INTRODUCTION

The Fourier Transform (FT) is a powerful analytical tool and has proved to be invaluable in many disciplines such as physics, mathematics and engineering. The development of the Fast Fourier Transform (FFT) algorithm (*Cooley & Tukey, 1965*), which computes the Discrete Fourier Transform (DFT) with a fast algorithm, firmly established the FT as a practical tool in diverse areas, most notably signal and image processing.

In two dimensions, the FFT can still be used to compute the DFT in Cartesian coordinates. However, in many applications such as photoacoustics (*Xu, Feng & Wang, 2002*) and tomography (*Scott et al., 2012*; *Fahimian et al., 2013*; *Lee et al., 2008*; *Lewitt & Matej, 2003*), it is often necessary to compute the FT in polar coordinates. Moreover, for functions that are naturally described in polar coordinates, a discrete version of the 2D FT in polar coordinates is needed. There have been some attempts to calculate the FT in polar coordinates, most notably through the Hankel transform, since the zeroth order

Corresponding author
Natalie Baddour,
nbaddour@uottawa.ca

Hankel transform is known to be a 2D FT in polar coordinates for rotationally symmetric functions. However, prior work has focused on numerically approximating the continuous transform. This stands in contrast to the FT, where the DFT can stand alone as an orthogonal transform, independent of the existence of its continuous counterpart.

The idea of a Polar FT has been previously investigated, where the spatial function is in Cartesian coordinates but its FT is computed in polar coordinates (*Averbuch et al., 2006*; *Abbas, Sun & Foroosh, 2017*; *Fenn, Kunis & Potts, 2007*). FTs have been proposed for non-equispaced data, referred to as Unequally Spaced FFT (USFFT) or Non-Uniform FFT (NUFFT) (*Dutt & Rokhlin, 1993*; *Fourmont, 2003*; *Dutt & Rokhlin, 1995*; *Potts, Steidl & Tasche, 2001*; *Fessler & Sutton, 2003*). A recent book gives a unified treatment of these topics (*Plonka et al., 2018*). Previous work has also considered the implications of using a polar grid (*Stark, 1979*; *Stark & Wengrovitz, 1983*). Although the above references demonstrate that the computation of a discrete 2D FT on a polar grid has previously been considered in the literature, there is to date no discrete 2D FT in polar coordinates that exists as a transform in its own right, with its own set of rules of the actual manipulated quantities.

In part I of this two part series, we proposed an independent discrete 2D FT in polar coordinates, which has been defined to be discrete from first principles (*Baddour, 2019*). For a discrete transform, the values of the transform are only given as entries in a vector or matrix and the transform manipulates a set of discrete values. To quote Bracewell (*Bracewell, 1999*), "we often think of this as though an underlying function of a continuous variable really exists and we are approximating it. From an operational viewpoint, however, it is irrelevant to talk about the existence of values other than those given and those computed (the input and output). Therefore, it is desirable to have a mathematical theory of the actual quantities manipulated". Hence, in our previous paper (*Baddour, 2019*), standard operational 'rules' of shift, modulation and convolution rules for this 2D DFT in polar coordinates were demonstrated. The operational rules were demonstrated via the key properties of the proposed discrete kernel of the transform. However, using the discrete kernel may not be the most effective way to compute the transform. Furthermore, while the 2D DFT in polar coordinates was demonstrated to have properties and rules as a standalone transform independent of its relationship to any continuous transform, an obvious application of the proposed discrete transform is to approximate its continuous counterpart.

Hence, the goal of this second part of this two-part paper series is to propose computational approaches to the computation of the previously proposed 2D DFT in polar coordinates and also to validate its effectiveness to approximate the continuous 2D FT in polar coordinates.

The outline of the paper is as follows. "Definition of the Discrete 2D FT in Polar Coordinates" states the proposed definition of the discrete 2D FT in polar coordinates. The motivation of this definition and the transform rules (multiplication, convolution,

shift etc.) are given in the first part of this two-part paper. The transform exists in its own right and manipulates discrete quantities that do not necessarily stem from sampling an underlying continuous quantity. Nevertheless, the motivation for the definition of the transform is based on an implied underlying discretization scheme. "Discrete Transform to approximate the continuous transform" introduces the implied underlying discretization scheme where we show the connection between discrete samples of the continuous functions and the discrete transform, should it be desirable to interpret the transform in this manner. Here, the connection between using the proposed 2D DFT and sampled vales of the continuous functions is explained. The proposed 2D DFT was motivated by a specific sampling scheme (Discrete Transform to approximate the continuous transform) which can be plotted and analyzed for "grid coverage"—how much of the 2D plane is covered and at which density. Thus, "Discretization Points and Sampling Grid" analyzes the proposed discretization points and their implication on the sampling grid for density and coverage of the grid. The insights gained from this section will be useful in interpreting the results of approximating the continuous transform with the discrete transform. "Numerical Computation of the Transform" introduces numerical computation schemes whereby the interpretation of the proposed 2D transform as a sequence of 1D DFT, 1D Discrete Hankel Transform (DHT) and 1D inverse DFT (IDFT) is exploited. "Numerical Evaluation of the 2D DFT in Polar Coordinates to Approximate the Continuous FT" then investigates the ability of the proposed 2D DFT to approximate the continuous transform in terms of precision and accuracy. Three test functions for which closed-form continuous transforms are known are analyzed. Finally, "Summary and Conclusion" summarizes and concludes the paper.

## DEFINITION OF THE DISCRETE 2D FT IN POLAR COORDINATES

The 2D-DFT in polar coordinates has been defined in the first part of this two-paper series as the discrete transform that takes the matrix (or double-subscripted series) $f_{pk}$ to the matrix (double-subscripted series) $F_{ql}$ such that $f_{pk} \rightarrow F_{qm}$ is given by

$$F_{qm} = \mathbb{F}(f_{pk}) = \sum_{k=1}^{N_1-1} \sum_{p=-M}^{M} f_{pk} E^{-}_{qm;pk} \tag{1}$$

where $p, k, q, m, n, N_1$, and $N_2$ are integers such that $-M \leq n \leq M$, where $2M + 1 = N_2$ $1 \leq m, k, \leq N_1 - 1$ and $-M \leq p, q \leq M$. Unless otherwise stated, in the remainder of the paper it shall be assumed that $p, k, q, m, n, N_1$, and $N_2$ are within these stated ranges. Similarly, for the inverse transform we propose

$$f_{pk} = \mathbb{F}^{-1}(F_{qm}) = \sum_{m=1}^{N_1-1} \sum_{q=-M}^{M} F_{qm} E^{+}_{qm;pk} \tag{2}$$

In Eqs. (1) and (2), $E^{\pm}_{qm;pk}$ are the kernels of the transformation. These can be chosen as the "non-symmetric" form given by

$$E^{-}_{qm;pk} = \frac{1}{N_2} \sum_{n=-M}^{M} \frac{J_n\left(\frac{j_{nk}j_{nm}}{j_{nN_1}}\right)}{j^2_{nN_1}J^2_{n+1}(j_{nk})} 2i^{-n} e^{-i\frac{2\pi np}{N_2}} e^{+i\frac{2\pi nq}{N_2}}$$

$$E^{+}_{qm;pk} = \frac{1}{N_2} \sum_{n=-M}^{M} \frac{J_n\left(\frac{j_{nm}j_{nk}}{j_{nN_1}}\right)}{J^2_{n+1}(j_{nm})} 2i^{+n} e^{+i\frac{2\pi np}{N_2}} e^{-i\frac{2\pi nq}{N_2}}$$

(3)

Here, $J_n(z)$ is the $n$th order Bessel function of the first kind and $j_{nk}$ denotes the $k$th zero of the $n$th Bessel function. The subscript (+ or −) indicated the sign on the $i^{\pm}$ and on the exponent containing the $p$ variable; the $q$ variable exponent then takes the opposite sign. From a matrix point of view, both $f_{pk}$ and $F_{ql}$ are $N_2 \times (N_1 - 1)$ sized matrices. The form of the kernel in Eq. (3) arises naturally from discretization of the continuous transform, but does not lead to the expected Parseval relationship. A possible symmetric kernel is discussed in the first part of this two-part paper and Parseval relationships are discussed further there (*Baddour, 2019*).

## DISCRETE TRANSFORM TO APPROXIMATE THE CONTINUOUS TRANSFORM

In this section, relationships between discretely sampled values of the function and its continuous 2D FT are presented in the case of a space-limited or band-limited function. These relationships were derived in the first part of the paper and are repeated here to demonstrate how they form the basis for the using the discrete transform to approximate the continuous transform at specified sampling points.

### Space-limited functions

Consider a function in the space domain $f(r, \theta)$ which is space limited to $r \in [0, R]$. This implies that the function is zero outside of the circle bounded by $r \in [0, R]$. An approximate relationship between sampled values of the continuous function and sampled values of its continuous forward 2D transform $F(\rho, \psi)$ has been derived in the first part of the two-part paper as

$$F\left(\frac{j_{qm}}{R}, \frac{2\pi q}{N_2}\right) \approx 2\pi R^2 \sum_{k=1}^{N_1-1} \sum_{p=-M}^{M} f\left(\frac{j_{pk}R}{j_{pN_1}}, \frac{2\pi p}{N_2}\right) \frac{1}{N_2} \sum_{n=-M}^{M} \frac{2i^{-n}J_n\left(\frac{j_{nk}j_{nm}}{j_{nN_1}}\right)}{j^2_{nN_1}J^2_{n+1}(j_{nk})} e^{-i\frac{2\pi np}{N_2}} e^{+i\frac{2\pi nq}{N_2}}$$

(4)

Similarly, an approximate relationship between sampled values of the continuous forward transform $F(\rho, \psi)$ and sampled values of the continuous original function $f(r, \theta)$ was shown to be given by

$$f\left(\frac{j_{pk}R}{j_{pN_1}}, \frac{2\pi p}{N_2}\right) \approx \frac{1}{2\pi R^2} \sum_{m=1}^{N_1-1} \sum_{q=-M}^{M} F\left(\frac{j_{qm}}{R}, \frac{2\pi q}{N_2}\right) \frac{1}{N_2} \sum_{n=-M}^{M} \frac{2i^{n}J_n\left(\frac{j_{nm}j_{nk}}{j_{nN_1}}\right)}{J^2_{n+1}(j_{nm})} e^{+i\frac{2\pi np}{N_2}} e^{-i\frac{2\pi nq}{N_2}}$$

(5)

In Eqs. (4) and (5), $f(r, \theta)$ is the original function in 2D space and $F(\rho, \psi)$ is the 2D FT of the function in polar coordinates.

To evaluate if the 2D DFT as proposed in Eqs. (1) and (2) can be used to approximate sampled values of $f(r, \theta)$ and $F(\rho, \psi)$, the process is as follows. For the forward transform, we start with the continuous $f(r, \theta)$, evaluate it at the sampling points and then assign this value to $f_{pk}$ via

$$f_{pk} = f\left(\frac{j_{pk}R}{j_{pN1}}, \frac{2\pi p}{N_2}\right) \tag{6}$$

Then, $F_{qm}$ is calculated from the 2D DFT scaled by $2\pi R^2$, Eq. (1), that is

$$F_{qm} = 2\pi R^2 \mathbb{F}(f_{pk}) = 2\pi R^2 \sum_{k=1}^{N_1-1} \sum_{p=-M}^{M} f_{pk} E_{qm;pk}^{-} \tag{7}$$

The factor of $2\pi R^2$ is necessary so that the evaluation in Eq. (7) matches the expression in Eq. (4). To evaluate if the proposed 2D DFT can be used to approximate the continuous transform, the question becomes how well $F_{qm}$ calculated from the 2D DFT in Eq. (7) approximates $F\left(\frac{j_{qm}}{R}, \frac{2\pi q}{N_2}\right)$—the values of the continuous 2D FT evaluated on the sampling grid.

To evaluate the inverse 2D DFT, the process is similar. We start with the continuous $F(\rho, \psi)$, evaluate it at the sampling points and assign this value to $F_{qm}$ via

$$F_{qm} = F\left(\frac{j_{qm}}{R}, \frac{2\pi q}{N_2}\right) \tag{8}$$

Now, $f_{pk}$ is calculated from a scaled version of the inverse 2D DFT, Eq. (2) that is

$$f_{pk} = \frac{1}{2\pi R^2} \mathbb{F}^{-1}(F_{qm}) = \frac{1}{2\pi R^2} \sum_{m=1}^{N_1-1} \sum_{q=-M}^{M} F_{qm} E_{qm;pk}^{+} \tag{9}$$

To evaluate if the proposed transform can approximate the continuous transform, the question becomes how well $f_{pk}$ calculated from Eq. (9) approximates $f\left(\frac{j_{pk}R}{j_{pN1}}, \frac{2\pi p}{N_2}\right)$—the values of the continuous function evaluated on the sampling grid.

## Band-limited functions

The process for band-limited functions follows the same process as outlined in the previous section, with the exception that the sampling points and scaling factors are slightly different as they are now given in terms of the band limit rather than the space limit. Now consider functions in the frequency domain $F(\rho, \psi)$ with an effective band limit $\rho \in [0, W_\rho]$. That is, we suppose that the 2D FT $F(\rho, \psi)$ of $f(r, \theta)$ is band-limited, meaning that $F(\rho, \psi)$ is zero for $\rho \geq W_\rho = 2\pi W$. The variable $W_\rho$ is written in this form since $W$ would typically be quoted in units of Hz (cycles per second) if using temporal units or cycles per meter if using spatial units. Therefore, the multiplication by $2\pi$ ensures that the final units are in $s^{-1}$ or $m^{-1}$. The approximate relationship between sampled

values of the continuous 2D FT $F(\rho, \psi)$ and sampled values of the original continuous function $f(r, \theta)$ was derived in the first part of the paper and is given by

$$F\left(\frac{j_{qm}W_\rho}{j_{qN_1}}, \frac{2\pi q}{N_2}\right) \approx \frac{2\pi}{W_\rho^2}\sum_{k=1}^{N_1-1}\sum_{p=-M}^{M} f\left(\frac{j_{pk}}{W_\rho}, \frac{2\pi p}{N_2}\right) \frac{1}{N_2}\sum_{n=-M}^{M} \frac{2i^{-n}J_n\left(\frac{j_{nm}j_{nk}}{j_{nN_1}}\right)}{J_{n+1}^2(j_{nk})} e^{-i\frac{2\pi np}{N_2}} e^{+i\frac{2\pi nq}{N_2}} \quad (10)$$

Similarly, the inverse relationship between sampled values of $F(\rho, \psi)$ and sampled values of $f(r, \theta)$ was shown to be given by

$$f\left(\frac{j_{pk}}{W_\rho}, \frac{2\pi p}{N_2}\right) \approx \frac{W_\rho^2}{2\pi}\sum_{m=1}^{N_1-1}\sum_{q=-M}^{M} F\left(\frac{j_{qm}W_\rho}{j_{qN_1}}, \frac{2\pi q}{N_2}\right) \frac{1}{N_2}\sum_{n=-M}^{M} \frac{2i^{n}J_n\left(\frac{j_{nk}j_{nm}}{j_{nN_1}}\right)}{j_{nN_1}^2 J_{n+1}^2(j_{nm})} e^{-i\frac{2\pi nq}{N_2}} e^{+i\frac{2\pi np}{N_2}} \quad (11)$$

The relationships in Eqs. (10) and (11) give relationships between the sampled values of the original function and sampled values of its 2D FT.

To evaluate the forward 2D DFT, we start with $f(r, \theta)$, evaluate it at the (bandlimited specific) sampling points and assign this value to $f_{pk}$ via

$$f_{pk} = f\left(\frac{j_{pk}}{W_\rho}, \frac{2\pi p}{N_2}\right) \quad (12)$$

Then, $F_{qm}$ is calculated from the discrete transform scaled by $\frac{2\pi}{W_\rho^2}$, Eq. (1), that is

$$F_{qm} = \frac{2\pi}{W_\rho^2}\mathbb{F}(f_{pk}) = \frac{2\pi}{W_\rho^2}\sum_{k=1}^{N_1-1}\sum_{p=-M}^{M} f_{pk}E_{qm;pk}^{-} \quad (13)$$

To evaluate if the proposed 2D DFT can be used to approximate the continuous transform, the question is how well $F_{qm}$ calculated from Eq. (13) approximates $F\left(\frac{j_{qm}W_\rho}{j_{qN_1}}, \frac{2\pi q}{N_2}\right)$, which are the values of the continuous 2D FT, evaluated on the sampling grid. The evaluation of the inverse transform for the band-limited function proceeds similarly by comparing values obtained from the inverse 2D DFT to the values obtained by sampling the continuous function directly.

The relationships given by Eqs. (4), (5), (10) and (11), were the motivating definition of a 2D DFT in polar coordinates, defined in the first part of this two-part paper. In the context of this second part of the two-part paper, they are also the relationships that permit the use of the discrete transform to approximate the continuous transform at the specified sampling points. They are also the relationships that permit the examination of whether the discrete quantities $f_{pk}$ and $F_{qm}$ calculated via the proposed 2D DFT are in fact reasonable approximations to the sampled values of the continuous functions, as stated in the objectives of the paper.

## DISCRETIZATION POINTS AND SAMPLING GRID

The transforms defined in Eqs. (1) and (2) can be applied to any matrix $f_{pk}$ to yield its forward transform $F_{qm}$, which can then be transformed backwards by using the inverse transform. However, if these same discrete transforms are to be used for the purpose of approximating a continuous 2D FT, then these transforms need to be applied to the

specific sampled values of the continuous functions in both space and frequency domains, as shown in Eqs. (6), (8) and (12). The relationships in Eqs. (4) and (10) define the sampling points that need to be used and it is noted that the points are defined differently based on whether we start with the assumption of a space or band limited function. These specific sampling points imply a specific sampling grid for the function. In this section, the sampling grid (its coverage and density in 2D) is analyzed.

## Sampling points

For a space-limited function, we assume that the original function of interest is defined over continuous $(r, \theta)$ space where $0 \leq r \leq R$ and $0 \leq \theta \leq 2\pi$. The discrete sampling spaces used for radial and angular sampling points in regular $\vec{r}$ space $(r, \theta)$ and $\vec{\omega}$ frequency $(\rho, \psi)$ space are defined as

$$r_{pk} = \frac{j_{pk}R}{j_{pN_1}} \qquad \theta_p = \frac{p2\pi}{N_2} \tag{14}$$

and

$$\rho_{qm} = \frac{j_{qm}}{R} \qquad \psi_q = \frac{q2\pi}{N_2} \tag{15}$$

For a band limited function, the function is assumed band-limited to $0 \leq \rho \leq W_\rho$, $0 \leq \psi \leq 2\pi$. The sampling space used for radial and angular sampling points in regular $\vec{\omega}$ frequency space $(\rho, \psi)$ and $\vec{r}$ space $(r, \theta)$ for a bandlimited function is defined as

$$r_{pk} = \frac{j_{pk}}{W_\rho} \qquad \theta_p = \frac{p2\pi}{N_2} \tag{16}$$

and

$$\rho_{qm} = \frac{j_{qm}W_\rho}{j_{qN_1}} \qquad \psi_q = \frac{q2\pi}{N_2} \tag{17}$$

Clearly, the density of the sampling points depends on the numbers of points chosen, that is on $N_1$ and $N_2$. Also clear is the fact that the grid is not equispaced in the radial variable. The sampling grid for a space-limited function are plotted below to enable visualization. In the first instance, the polar grids are plotted for the case $R = 1$, $N_1 = 16$ and $N_2 = 15$. These are shown in space ($\boldsymbol{r}$ space) and frequency ($\boldsymbol{\rho}$ space) in Figs. 1 and 2 respectively. It should be noted that although we refer the grids in this article as polar grids, they are not true polar grids in the sense of equispaced sampling in the radial and angular coordinates.

Clearly, the grids in Figs. 1 and 2 are fairly sparse, but the low values of $N_2$ and $N_1$ have been chosen so that the structure of the sampling points can be easily seen. It can be observed that there is a hole at the center area in both domains which is caused by the special sampling points. For higher values of the $N_2$ and $N_1$, the grid becomes fairly dense, obtaining good coverage of both spaces, but details are harder to observe. To demonstrate,

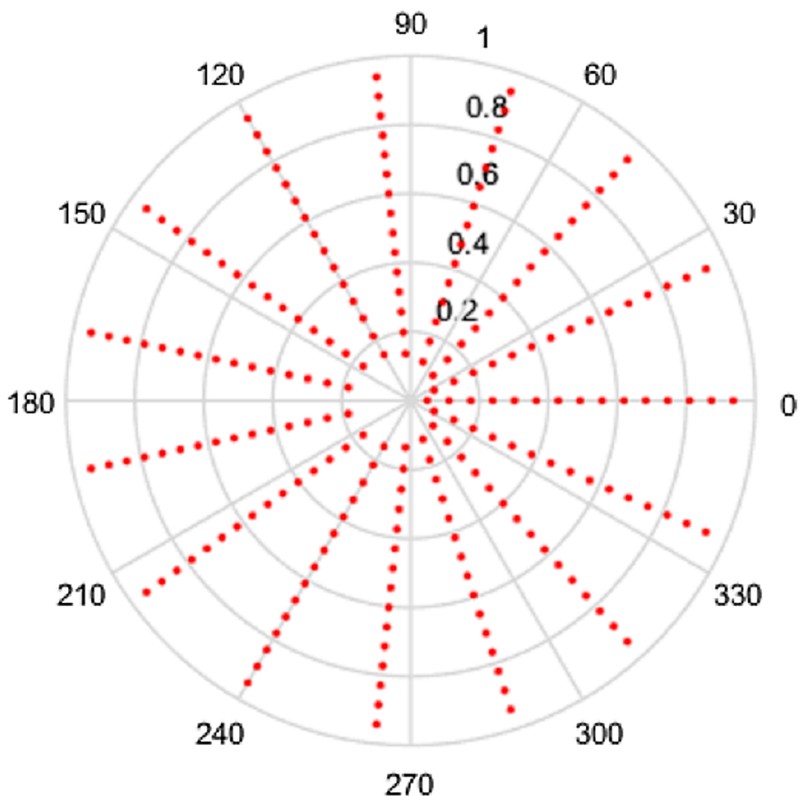

**Figure 1 Spatial sampling grid for a space-limited function with $R = 1$, $N_1 = 16$ and $N_2 = 15$.**

the polar grids are plotted for the case $R = 1$, $N_1 = 96$ and $N_2 = 95$. These are shown in Figs. 3 and 4 respectively.

From Figs. 3 and 4, by choosing higher values of $N_1$ and $N_2$, the sampling grid becomes denser, however there is still a gap in the center area. The sampling grids for band-limited functions are not plotted here since the sample grid for a band-limited function has the same shape as with space limited function but the domains are reversed.

## Sample grid analysis

From part I of the paper, it was shown that the 2D-FT can be interpreted as a DFT in the angular direction, a DHT in the radial direction and then an IDFT in the angular direction. Hence, the sample size in the angular direction could have been decided by the Nyquist sampling theorem (*Shannon, 1984*), which states that

$$f_s > 2f_{max} \tag{18}$$

where $f_s$ is the sample frequency and $f_{max}$ is the highest frequency or band limit.

In the radial direction, the necessary relationship for the DHT is given by *Baddour & Chouinard (2015)*

$$W_\rho R = j_{nN_1} \tag{19}$$

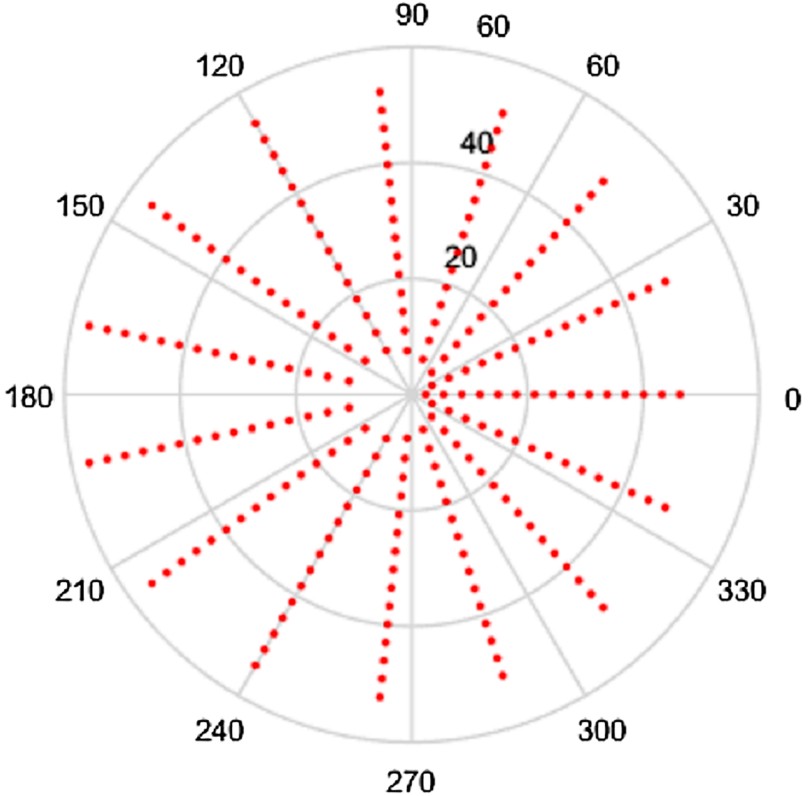

**Figure 2 Frequency space sampling grid for a space-limited function with $R = 1$, $N_1 = 16$ and $N_2 = 15$.**

where $W_\rho$ is the effective band-limit, $R$ is the effective space limit and $j_{nN}$ is the $N$th zero of $J_n(r)$. For the 2D FT, since $-M \leq p \leq M$, the order of the Bessel zero ranges from $-M$ to $M$, the required relationship becomes

$$\min(j_{pN_1}) \geq W_\rho R \qquad (20)$$

The relationships $j_{nN} = j_{-nN}$ and $j_{0N_1} < j_{\pm 1N_1} < j_{\pm 2N_1} < \dots < j_{\pm MN_1}$ are valid (*Lozier, 2003*), hence Eq. (20) can be written as

$$j_{0N_1} \geq W_\rho R \qquad (21)$$

It is pointed out in *Baddour (2019)* and *Guizar-Sicairos & Gutiérrez-Vega (2004)* that the zeros of $J_n(z)$ are almost evenly spaced at intervals of $\pi$ and that the spacing becomes exactly $\pi$ in the limit as $z \to \infty$. The reader unfamiliar with Bessel functions is directed to references (*Bracewell, 1999*; *Lozier, 2003*). In fact, it is shown in *Dutt & Rokhlin (1993)* that a simple asymptotic form for the Bessel function is given by

$$J_n(z) \approx \sqrt{\frac{2}{\pi z}} \cos\left[z - \left(n + \frac{1}{2}\right)\frac{\pi}{2}\right] \qquad (22)$$

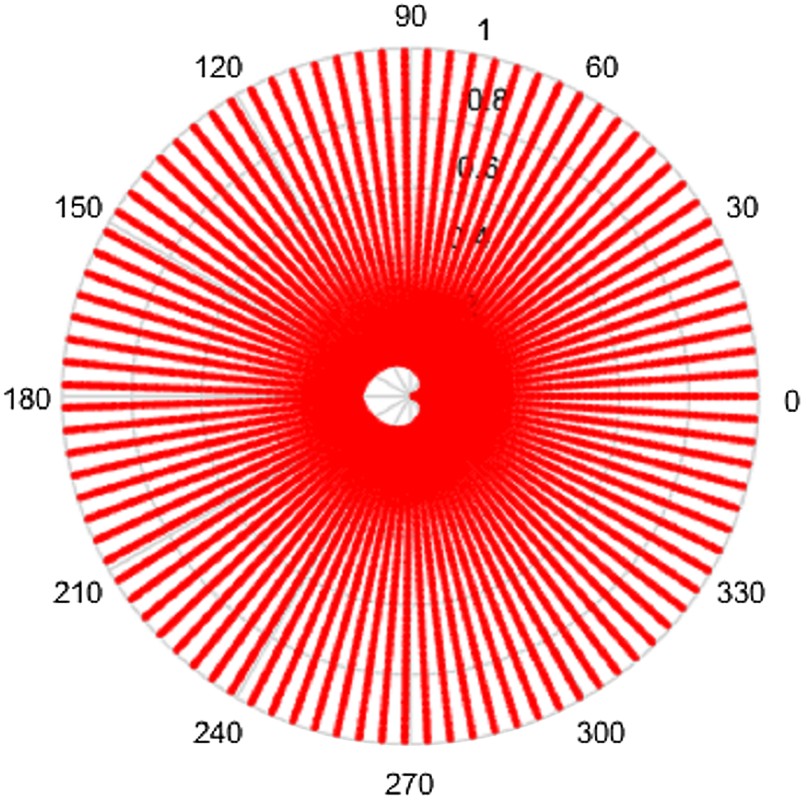

**Figure 3 Spatial sampling grid for a space-limited function with $R = 1$, $N_1 = 96$ and $N_2 = 95$.**

Therefore, an approximation to the Bessel zero, $j_{nk}$ is given by

$$j_{nk} \approx \left(2k + n - \frac{1}{2}\right)\frac{\pi}{2} \tag{23}$$

Hence, Eq. (21) can be written to choose $N_1$ approximately as

$$N_1 \pi \geq W_\rho R = 2\pi W R$$
$$\Rightarrow N_1 \geq 2WR \tag{24}$$

where the reader is reminded that the units of $W$ is m$^{-1}$ (the space equivalent of Hz). $N_1/R$ is the spatial sampling frequency and we see that Eq. (24) effectively makes the same statement as Eq. (18), as it should.

Intuitively, more sample points lead to more information captured, which gives an expectation that increasing $N_1$ or $N_2$ individually will give a better sampling grid coverage. However, it can be seen from Figs. 1–4 that there is a gap in the center of the sample grid. From Eqs. (14) and (15), the area of the gap in the center is related to the ranges of $p$ and $k$, that is $N_2$ and $N_1$. In the sections below, it is assumed that the sampling theorems are already satisfied (that is, an appropriate space and band limit is selected) and the relationship between $N_2$, $N_1$ and the size of the gap will be discussed.

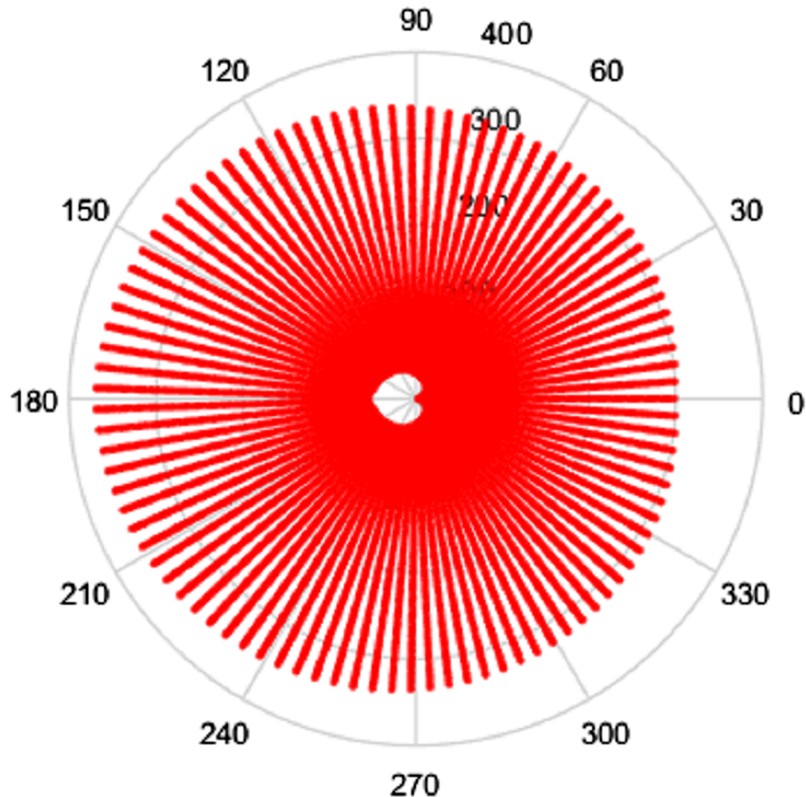

**Figure 4 Frequency space sampling grid for a space-limited function with $R = 1$, $N_1 = 96$ and $N_2 = 95$.**

### Space-limited function

In this section, it is assumed that the function is a space limited function, defined in
$r \in [0, R]$. The sampling points are defined as Eq. (14) in the space domain and Eq. (15) in
the frequency domain. In the following, a relationship between $N_2$, $N_1$ and the area of the
gap in both domains is discussed.

### Sample grid in the space domain

In the space domain, the effective limit in the space domain, $R$, is fixed. To analyze how the
values of $N_2$ and $N_1$ affect the coverage of the grid in space domain, consider the following
definition of 'grid coverage'

$$A_r = \frac{\pi R^2 - \pi \bar{r}^2}{\pi R^2} \cdot 100 \tag{25}$$

where $\bar{r}$ denotes the average radius of the gap (the hole in the middle of the grid). $A_r$ as
defined in Eq. (25) is a measure of the "grid coverage" since it gives a percentage of how
much of the original space limited domain area is captured by the discrete grid. For
example, if the average radius of the center gap is zero, then $A_r$ would be 100%, that is,
complete coverage. Based on the observation of Figs. 1 and 3, the relationship

**Table 1 Spatial grid coverage, $A_r$, with respect to different values of $N_1$ and $N_2$ ($R$ is fixed).**

| $N_2$ | $N_1$ | | | |
|---|---|---|---|---|
| | 15 | 75 | 150 | 300 |
| 15 | $A_r = 98.48\%$ | $A_r = 99.92\%$ | $A_r = 99.98\%$ | $A_r = 99.99\%$ |
| 75 | $A_r = 93.78\%$ | $A_r = 99.36\%$ | $A_r = 99.81\%$ | $A_r = 99.95\%$ |
| 151 | $A_r = 90.14\%$ | $A_r = 98.42\%$ | $A_r = 99.46\%$ | $A_r = 99.84\%$ |
| 301 | $A_r = 86.17\%$ | $A_r = 96.58\%$ | $A_r = 98.59\%$ | $A_r = 99.51\%$ |

$r_{01} < r_{\pm 11} < r_{\pm 21} < r_{\pm M1}$ is valid. Therefore, from Eq. (14), the average radius of the gap is given by

$$\bar{r} = \frac{(r_{01} + r_{M1})}{2} = \frac{1}{2}\left(\frac{j_{01}}{j_{0N_1}}R + \frac{j_{M1}}{j_{MN_1}}R\right) \tag{26}$$

Hence, Eq. (25) for grid coverage can be written as

$$A_r = \left[1 - \frac{1}{4}\left(\frac{j_{01}}{j_{0N_1}} + \frac{j_{M1}}{j_{MN_1}}\right)^2\right]\cdot 100 \tag{27}$$

Table 1 shows the different values of grid coverage $A_r$ as the values of $N_1$ and $N_2$ are changed.

From Table 1, it can be seen that increasing $N_1$ (sample size in the radial direction) tends to increase the grid coverage. Since the effective space limit $R$ is fixed, from Eq. (21) it follows that increasing $N_1$ actually increases the effective band limit. However, increasing $N_2$ (sample size in angular direction) will result in a bigger gap in the center of the grid, which then decreases the coverage.

*Sample grid in the frequency domain*

Similarly, coverage of the grid in the frequency domain is defined as

$$A_\rho = \frac{\pi W_\rho^2 - \pi\bar{\rho}^2}{\pi W_\rho^2}\cdot 100 \tag{28}$$

where $\bar{\rho}$ denotes the average radius of the gap. Since

$$\bar{\rho} = \frac{(\rho_{01} + \rho_{M1})}{2} = \frac{(j_{01} + j_{M1})}{2R} \tag{29}$$

Then, it follows that Eq. (28) for frequency domain grid coverage can be written as

$$A_\rho = \left[1 - \frac{(j_{01} + j_{M1})^2}{4R^2 W_\rho^2}\right]\cdot 100\% \tag{30}$$

From Eq. (30), it can be observed that the sample grid coverage in the frequency domain is affected by $R$, $W_\rho$ and $M$. Since $N_2 = 2M + 1$, in order to get a better grid coverage with a fixed $W_\rho$, $R$ and $N_2$ can be adjusted. Table 2 shows the grid coverage $A_\rho$ for different values of $R$ and $N_2$.

From Table 2, the conclusion for the frequency domain is that when the effective band limit is fixed, increasing $R$ (effective space limit) tends to increase the coverage in the

**Table 2 Frequency grid coverage, $A_\rho$, with respect to different values of $R$ and $N_2$ ($W_\rho$ is fixed).**

| $N_2$ | R | | | |
|---|---|---|---|---|
| | 15 | 75 | 150 | 300 |
| 15 | $A_\rho$ = 99.80% | $A_\rho$ = 99.99% | $A_\rho$ = 100.00% | $A_\rho$ = 100.00% |
| 75 | $A_\rho$ = 97.66% | $A_\rho$ = 99.91% | $A_\rho$ = 99.98% | $A_\rho$ = 99.99% |
| 151 | $A_\rho$ = 91.88% | $A_\rho$ = 99.68% | $A_\rho$ = 99.92% | $A_\rho$ = 99.98% |
| 301 | $A_\rho$ = 70.67% | $A_\rho$ = 98.83% | $A_\rho$ = 99.71% | $A_\rho$ = 99.93% |

frequency domain, while increasing $N_2$ (sample size in the angular direction) decreases the coverage. However, from Eq. (21) it should be noted that to satisfy the sampling theorem, increasing $R$ with fixed $W_\rho$ requires an increase in $N_1$ at the same time.

### Band-limited function

In this section, we suppose that the function is an effectively band limited function, defined on $\rho \in [0, W_p]$. The sampling points are defined as in Eq. (16) in the space domain and as in the frequency domain. In this subsection, the relationship between $N_2$, $N_1$ and the area of the gap in both domains is discussed.

#### Sampling grid in the space domain

The same definition of grid coverage in the space domain will be used as in Eq. (25). Since the sampling points of a band-limited function are given by Eqs. (16) and (17), the average radius of the gap can be defined as

$$\bar{r} = \frac{(r_{01} + r_{M1})}{2} = \frac{1}{2}\left(\frac{j_{01}}{W_\rho} + \frac{j_{M1}}{W_\rho}\right) \tag{31}$$

Therefore, the coverage of the grid in space domain can be written as

$$A_r = \left[1 - \frac{(j_{01} + j_{M1})^2}{4W_\rho^2 R^2}\right] \cdot 100 \tag{32}$$

It can be observed that the grid coverage in the space domain of a band-limited function is the same as the grid coverage in the frequency domain of space limited function.

#### Sample grid in frequency domain

The coverage of the grid in the frequency domain of a band limited function is defined by Eq. (28). With sampling points defined in Eq. (17), the average radius of the gap can be defined as

$$\bar{\rho} = \frac{(\rho_{01} + \rho_{M1})}{2} = \frac{1}{2}\left(\frac{j_{01}}{j_{0N_1}}W_\rho + \frac{j_{M1}}{j_{MN_1}}W_\rho\right) \tag{33}$$

The coverage of the grid in frequency domain can be written as

$$A_\rho = \left[1 - \frac{1}{4}\left(\frac{j_{01}}{j_{0N_1}} + \frac{j_{M1}}{j_{MN_1}}\right)^2\right] \cdot 100 \tag{34}$$

It can be observed that the grid coverage in the frequency domain of a band-limited function is the same as the grid coverage in the space domain of a space limited function.

## Conclusion

Based on the discussion above, the following conclusions can be made:

1. Increasing $N_2$ (angular direction) tends to decrease the sampling grid coverage in both domains. Increasing $N_1$ (radial direction) tends to increase the sampling coverage in the space domain for a space-limited function and in the frequency domain for a frequency-limited function. So, if a signal changes sharply in the angular direction such that large values of $N_2$ are needed, a large value of $N_1$ is also needed to compensate for the effect of increasing $N_2$ on the grid coverage.

2. For a space-limited function, if there is a lot of energy at the origin in the space domain, a larger value of $N_1$ will be required to ensure that the sampling grid gets as close to the origin as possible in the space domain. If the function has a lot of energy at the origin in the frequency domain, a large value for both $N_1$ and $R$ will be required to ensure adequate grid coverage.

3. For a band-limited function, if there is a lot of energy at the origin in the frequency domain, a large value of $N_1$ will be needed to ensure that the sample grid gets as close to the origin as possible in the frequency domain. If the function has a lot of energy at the origin in the space domain, large values for both $N_1$ and $W_\rho$ are required.

## NUMERICAL COMPUTATION OF THE TRANSFORM

We have already demonstrated in part I of the paper that the discrete 2D FT in polar coordinates can be interpreted as a DFT, DHT and then IDFT. This interpretation is quite helpful in coding the transform and in exploiting the speed of the FFT (Fast Fourier Transform) in implementing the computations. In this section, we explain how the speed of Matlab's (Mathworks 2018) built-in code (or similar software) can be exploited to implement the 2D DFT in polar coordinates.

### Forward transform

The values $f_{pk}$ can be considered as the entries in a matrix. To transform $f_{pk} \rightarrow F_{qm}$, the operation is performed as a sequence of steps which are a 1D DFT (column-wise), followed by a scaled 1D DHT (row-wise), finally followed by a 1D IDFT (column-wise). The reader is reminded that the range of indices is given by $m, k = 1 \ldots N_1 - 1$ and $n, p, q = -M \ldots M$, where $2M + 1 = N_2$. These steps can be summarized succinctly by rewriting Eq. (1) as

$$
F_{qm} = \frac{1}{N_2} \sum_{n=-M}^{M} \left[ \underbrace{\frac{2\pi R^2 i^{-n}}{j_{nN_1}} \sum_{k=1}^{N_1-1} Y_{m,k}^{nN_1} \underbrace{\sum_{p=-M}^{M} f_{pk} e^{-in\frac{2\pi p}{N_2}}}_{\text{1D DFT column-wise}}}_{\text{scaled 1D DHT row-wise}} \right] e^{+in\frac{2\pi q}{N_2}} \tag{35}
$$

inverse 1D DFT column-wise

where the DHT is defined in *Baddour & Chouinard (2015)* via the transformation matrix

$$Y_{m,k}^{nN_1} = \frac{2}{j_{nN_1} J_{n+1}^2 (j_{nk})} J_n \left( \frac{j_{nm} j_{nk}}{j_{nN_1}} \right) \qquad 1 \le m, k \le N_1 - 1 \tag{36}$$

Matlab code for the DHT is described in *Baddour & Chouinard (2017)*. The inverse 2D DFT can be similarly interpreted, as shown in "Inverse Transform".

## Inverse transform

The steps of the inverse 2D DFT are the reverse of the steps outlined above for the forward 2D DFT. For $p = -M \dots M$ and $k = 1 \dots N_1 - 1$, Eq. (2) this can be expressed as

$$f_{pk} = \frac{1}{N_2} \sum_{n=-M}^{M} \left[ \underbrace{\frac{j_{nN_1} i^{+n}}{2\pi R^2} \sum_{m=1}^{N_1-1} Y_{k,m}^{nN_1} \underbrace{\sum_{q=M}^{M} F_{qm} e^{-i\frac{2\pi nq}{N_2}}}_{\text{1D DFT (column-wise)}}}_{\text{scaled 1D DFT (row-wise)}} \right] e^{+i\frac{2\pi np}{N_2}} \tag{37}$$

$$\underbrace{\phantom{XXXXXXXXXXXXXXXXXXXXXXXXXXXXXXXXXXXXXXXXXXXXX}}_{\text{inverse 1D DFT (column-wise)}}$$

This parallels the steps taken for the continuous case, with each continuous operation (Fourier series, Hankel transform) replaced by its discrete counterpart (DFT, DHT).

Therefore, for both forward and inverse 2D-DFT, the sequence of operations is a DFT of each column of the starting matrix, followed by a DHT of each row, a term-by-term scaling, followed by an IDFT of each column. This is a significant computational improvement because by interpreting the transform this way, the Fast Fourier Transform (FFT) can be used, which reduces the computational time quite significantly in comparison with a direct implementation of the summation definitions in Eqs. (1) and (2).

## Interpretation of the sampled forward transform in Matlab terms

To use the built-in Matlab function *fft*, a few operations are required. First, we define Matlab-friendly indices $p' = p + (M + 1)$ and $n' = n + (M + 1)$ so that $p, n = -M \dots M$ become $p', n' = 1 \dots 2M + 1 = 1 \dots N_2$ (since $2M + 1 = N_2$). That is, the primed variables range from $1 \dots 2M + 1$ rather than $-M \dots M$. Hence, if the matrix **f** with entries $f_{p'k}$ is defined, where $p' = 1 \dots N_2$, $k = 1 \dots N_1 - 1$, then the first step in which is a column-wise DFT can be written as the Matlab-defined DFT as

$$\bar{f}_{n'k} = \sum_{p'=1}^{N_2} f_{pk} e^{\frac{-2\pi i (p'-1-M)(n'-1-M)}{N_2}} \tag{38}$$

The overbar denotes a DFT. The definition of DFT in Matlab is actually given by the relationship

$$\bar{f}_{n'k} = \sum_{p'=1}^{N_2} f_{p'k} e^{\frac{-2\pi i (p'-1)(n'-1)}{N_2}} \tag{39}$$

Since the relationship $\sum_{p'=1}^{N_2} f_{p'k} e^{\frac{-2\pi i(p'-1)(n'-1-M)}{N_2}} = \sum_{p'=1}^{N_2} f_{pk} e^{\frac{-2\pi i(p'-1-M)(n'-1-M)}{N_2}}$ is valid, we can sample the original function to obtain the discrete $f_{pk}$ values, put them in the matrix $f_{p'k}$ then shift the matrix $f_{p'k}$ by $M+1$ along the column direction. In Matlab, the function $circshift(A, K, \dim)$ can be used, which circularly shifts the values in array $A$ by $K$ positions along dimension $\dim$. Inputs $K$ and $\dim$ must be scalars. Specifically, $\dim = 1$ indicates the columns of matrix $A$ and $\dim = 2$ indicates the rows of matrix $A$. Hence, Eq. (38) can be written as

$$\bar{f}_{n'k} = fft\big(circshift(f_{p'k}, M+1, 1), N_2, 1\big) \tag{40}$$

In matrix operations, this is equivalent to stating that each *column* of $f_{p'k}$ is DFT'ed to yield $\bar{f}_{n'k}$.

The second step in Eq. (35) is a DHT of order $n$, transforming $\bar{f}_{n'k} \to \hat{\bar{f}}_{n'l}$ so that the $k$ subscript is Hankel transformed to the $l$ subscript. The overhat denotes a DHT. In order to relate the order $n$ to the index $n'$, we need to shift $\bar{f}_{n'k}$ by $-(M+1)$ along column direction so that the order ranges from $-M$ to $M$.

$$\hat{\bar{f}}_{n'l} = \sum_{k=1}^{N_1-1} \frac{2J_n\left(\frac{j_{nk}j_{nl}}{j_{nN_1}}\right)}{j_{nN_1}J_{n+1}^2(j_{nk})} circshift\big(\bar{f}_{n'k}, -(M+1), 1\big) \quad \begin{cases} \text{for } n' = 1 \ldots N_2, \ l = 1 \ldots N_1 - 1 \\ \text{where } n = n' - (M+1) \end{cases} \tag{41}$$

By using the Hankel transform matrix defined in *Baddour & Chouinard (2015)*, Eq. (41) can be rewritten as

$$\hat{\bar{f}}_{n'l} = circshift\big(\bar{f}_{n'k}, -(M+1), 1\big) \left(Y_{l,k}^{nN_1}\right)^T \quad \begin{cases} \text{for } n' = 1 \ldots N_2, \ l = 1 \ldots N_1 - 1 \\ \text{where } n = n' - M - 1 \end{cases} \tag{42}$$

In matrix operations, this states that each *row* of $\bar{f}_{n'k}$ is DHT'ed to yield $\hat{\bar{f}}_{n'l}$. These are now scaled to give the Fourier coefficients of the 2D DFT $\hat{\bar{f}}_{n'l} \to \bar{F}_{n'l}$. In order to proceed to an IDFT in the next step, it is necessary to shift the matrix by $M+1$ along the column direction after scaling

$$\bar{F}_{n'l} = circshift\left(\frac{2\pi R^2}{j_{nN_1}} i^{-n} \hat{\bar{f}}_{n'l}, M+1, 1\right) \quad \begin{cases} \text{for } n' = 1 \ldots N_2, \ l = 1 \ldots N_1 - 1 \\ \text{where } n = n' - (M+1) \end{cases} \tag{43}$$

This last step is a 1D IDFT for each *column* of $\bar{F}_{n'l}$ to obtain $F_{ql}$. Using $2M+1 = N_2$, and $q' = q + 1 + M$, this can be written as

$$\begin{aligned}
F_{q'l} &= \frac{1}{N_2} \sum_{n'=1}^{N_2} \bar{F}_{nl} e^{+i(n'-M-1)\frac{2\pi(q'-1-M)}{N_2}} \quad \text{for } q' = 1 .. N_2, \quad l = 1 .. N_1 - 1 \\
&= \frac{1}{N_2} \sum_{n'=1}^{N_2} \bar{F}_{n'l} e^{+i(n'-1)\frac{2\pi(q'-1-M)}{N_2}} \\
&= circshift(ifft(\bar{F}_{n'l}, N_2, 1), -(M+1), 1)
\end{aligned} \tag{44}$$

### Interpretation of the sampled inverse transform in Matlab terms

Similar to the forward transform, matlab-friendly indices $q' = q + (M + 1)$ and $n' = n + (M + 1)$ are also defined. Hence, if the matrix $F$ with entries $F_{q'l}$ is defined, where $q' = 1 \ldots N_2,\ l = 1 \ldots N_1 - 1$, it then follows that the first 1D DFT step in Eq. (37) can be written as the Matlab-defined DFT as

$$
\bar{F}_{n'l} = \sum_{q'=1}^{N_2} F_{ql} e^{-i(n'-M-1)\frac{2\pi(q'-1-M)}{N_2}} \quad \text{for } n' = 1 \ldots N_2, \quad l = 1 \ldots N_1 - 1
$$

$$
= \sum_{q'=1}^{N_2} F_{q'l} e^{-i(n'-M-1)\frac{2\pi(q'-1)}{N_2}}
$$
(45)

If the original function can be sampled as $F_{ql}$ and then put into matrix $F_{q'l}$, then we need an *circshift* operation. So Eq. (45) can be written as

$$
\bar{F}_{n'l} = fft\left(circshift(F_{q'l}, M + 1, 1), N_2, 1\right)
$$
(46)

Subsequently, a DHT of order $n$ is required, transforming $\bar{F}_{n'l} \to \hat{\bar{F}}_{n'l}$ so that the $l$ subscript is Hankel transformed to the $k$ subscript. To achieve this, *circshift* is also needed here.

$$
\hat{\bar{F}}_{n'k} = circshift(\bar{F}_{n'l}, -(M + 1), 1)\left(Y_{k,l}^{nN_1}\right)^T \quad
\begin{cases}
\text{for } n' = 1 \ldots N_2, \ l = 1 \ldots N_1 - 1 \\
\text{where } n = n' - M - 1
\end{cases}
$$
(47)

This is followed by a scaling operation to obtain $\hat{\bar{F}}_{n'k} \to \bar{f}_{n'k}$ and then a *circshift* by $(M + 1)$ so that

$$
\bar{f}_{n'k} = circshift\left(\frac{j_{nN_1}}{2\pi R^2} i^{+n} \hat{\bar{F}}_{n'k}, (M + 1), 1\right) \quad
\begin{cases}
\text{for } n' = 1 \ldots N_2, \ k = 1 \ldots N_1 - 1 \\
\text{where } n = n' - (M + 1)
\end{cases}
$$
(48)

This last step is a 1D IDFT for each *column* of $\bar{f}_{n'k}$ to get $f_{p'k}$. Using $2M + 1 = N_2$, and $p' = p + 1$, Eq. (37) can be written as

$$
f_{p'k} = \frac{1}{N_2} \sum_{n'=1}^{N_2} \bar{f}_{nk} e^{+i(n'-M-1)\frac{2\pi(p'-1-M)}{N_2}} \quad \text{for } p' = 1 \ldots N_2, \quad k = 1 \ldots N_1 - 1
$$

$$
= \frac{1}{N_2} \sum_{n'=1}^{N_2} \bar{f}_{n'k} e^{+i\frac{2\pi(n'-1)(p'-1-M)}{N_2}}
$$
(49)

$$
= circshift\left(ifft\left(\bar{f}_{n'k}, N_2, 1\right), -(M + 1), 1\right)
$$

In conclusion, in this section, by using the interpretation of the kernel as sequential DFT, DHT and IDFT operations, Matlab (or similar software) built-in code can be used to efficiently implement the 2D DFT algorithm in polar coordinates.

## NUMERICAL EVALUATION OF THE 2D DFT IN POLAR COORDINATES TO APPROXIMATE THE CONTINUOUS FT

In this section, the 2D DFT is evaluated for its ability to estimate the continuous FT at the selected special sampling points in the spatial and frequency domains.

## Method for testing the algorithm
### Accuracy

In order to test accuracy of the 2D-DFT and 2D-IDFT to calculate approximate the continuous counterpart, the dynamic error is proposed as a metric. The dynamic error is defined as *Guizar-Sicairos & Gutiérrez-Vega (2004)*

$$E(v) = 20 \log_{10} \left[ \frac{|C(v) - D(v)|}{\max |D(v)|} \right] \tag{50}$$

where $C(v)$ is the continuous forward or inverse 2D-FT and $D(v)$ is the value obtained from the discrete counterpart. The dynamic error is defined as the ratio of the absolute error to the maximum amplitude of the discrete function, calculated on a log scale. Therefore, a large negative value represents an accurate discrete transform. The dynamic error is used instead of the percentage error in order to avoid division by zero.

### Precision

The precision of the algorithm is an important evaluation criterion, which is tested by sequentially performing a pair of forward and inverse transforms and comparing the result to the original function. High precision indicates that numerical evaluation of the transform does not add much error. An average of the absolute error between the original function and the calculated counterpart at each sample point is used to measure the precision. It is given by

$$\varepsilon = \frac{1}{(N_1 - 1) \cdot N_2} \sum_{n=1}^{(N_1 - 1) \cdot N_2} |f - f^*| \tag{51}$$

where $f$ is the original function and $f^*$ is the value obtained after sequentially performing a forward and then inverse transform. An ideal precision would result in the absolute error being zero.

## Test functions

In this section, three test functions are chosen to evaluate the ability of the discrete transform to approximate the continuous counterpart. The first test case is the circularly symmetric Gaussian function. Given that it is circularly symmetric and that the Gaussian is continuous and smooth, the proposed DFT is expected to perform well. The second test case is "Four-term sinusoid and Sinc" function, which is not symmetric in the angular direction and suffers a discontinuity in the radial direction. The third test function presents a more challenging test function, the "Four-term sinusoid and Modified exponential" function. In this case, the test function is not circularly symmetric and it explodes at the origin (approaches infinity at the origin). Given that as shown above, the sampling grid cannot cover the area around the origin very well, a function that explodes at the origin should give more error and should provide a reasonable test case for evaluating the performance of the discrete transform. The test functions are chosen to test specific aspects of the performance of the discrete transform but also because a closed-form expression for both the function and its transform are available. This then allows a

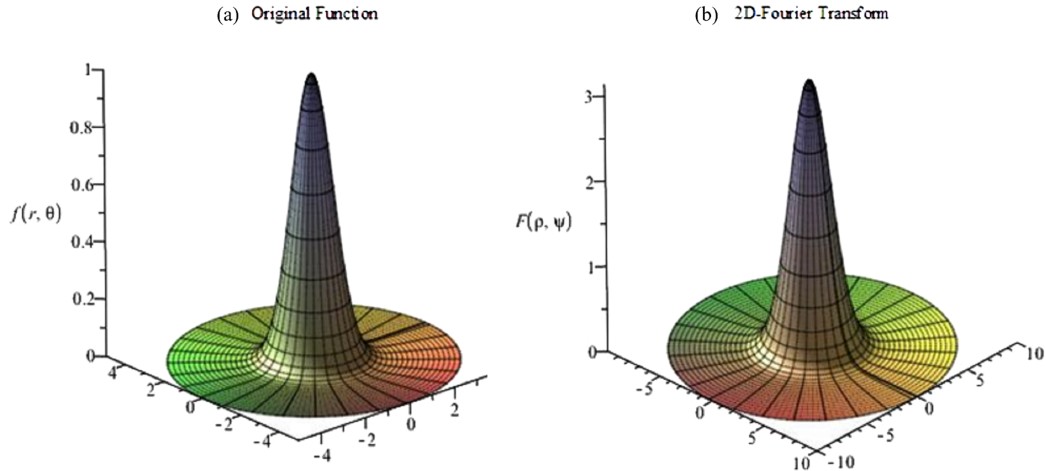

**Figure 5 (A)** Original function (Gaussian) and **(B)** its continuous 2D-DFT (which is also a Gaussian). 

numerical evaluation of the error between the quantities computed with the 2D DFT and the quantities obtained by evaluating (sampling) the continuous (forward or inverse) transform at the grid points.

### Gaussian

The first function chosen for evaluation is a circular symmetric function which is Gaussian in the radial direction. Specifically, the function in the space domain is given by

$$f(r, \theta) = e^{-a^2 r^2} \tag{52}$$

where $a$ is some real constant. Since the function is circularly symmetric, the 2D-DFT is a zeroth-order Hankel Transform (*Poularikas, 2010*) and is given by

$$F(\rho, \psi) = \frac{\pi}{a^2} e^{\frac{-\rho^2}{4a^2}} \tag{53}$$

The graphs for the original function and its continuous 2D-DFT (which is also a Gaussian) are plotted with $a = 1$ and shown in Fig. 5. From Fig. 5, the function is circular symmetric and fairly smooth in the radial direction. Moreover, the function can be considered as either an effectively space-limited function or an effectively band-limited function. For the purposes of testing it, it shall be considered as a space-limited function and Eqs. (14) and (15) will be used to proceed with the forward and inverse transform in sequence.

To perform the transform, the following variables need to be chosen: $N_2$, $R$ and $N_1$. In the angular direction, since the function in the spatial domain is circularly symmetric, $N_2$ can be chosen to be small. Thus, $N_2 = 15$ is chosen.

In the radial direction, from plotting the function, it can be seen that the effective space limit can be taken to be $R = 5$ and the effective band limit can be taken to be $W_\rho = 10$. From Eq. (21), $j_{0N_1} \geq R \cdot W_\rho = 50$. Therefore, $N_1 = 17$ is chosen (we could also have obtained a rough estimate of $N_1$ from Eq. (24)). However, most of the energy of the

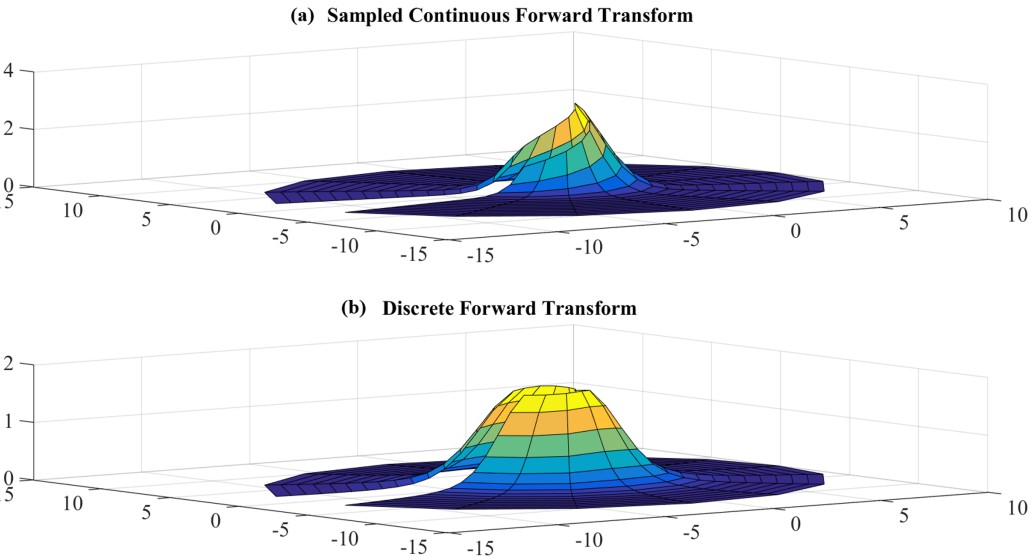

**Figure 6** (A) sampled continuous transform and (B) discrete forward transform for a Gaussian function with $R = 5$ and $N_1 = 17$.  

function in both the space and frequency domains is located in the center near the origin. Based on the discussion in "Conclusion", relatively large values of $R$ and $W_\rho$ are needed. The effective space limit $R = 40$ and effective band-limit $W_p = 30$ are thus chosen, which gives $j_{0N_1} \geq R \cdot W_\rho = 1200$. Therefore $N_1 = 383$ is chosen in order to satisfy this constraint. Both cases discussed here ($N_1 = 17$ and $N_1 = 383$) are tested in following.

*Forward transform*
Test results with $R = 5$, $N_1 = 17$ are shown in Figs. 6 and 7. Figure 6 shows the sampled continuous forward transform and the discrete forward transform. Figure 7 shows the error between the sampled values of the continuous transform and the discretely calculated values.

From Fig. 7, it can be observed that the error gets bigger at the center, which is as expected because the sampling grid shows that the sampling points can never attain the origin. The maximum value of the error is $E_{max} = -0.9115$ dB and this occurs at the center. The average error is $E_{avg.} = -30.4446$ dB.

Error test results with $R = 40$, $N_1 = 383$ are shown in Fig. 8. Similar to the previous case, the error gets larger at the center, as expected. However, the maximum value of the error is $E_{max} = -8.3842$ dB and this occurs at the center. The average value of the error is $E_{avg.} = -63.8031$ dB. Clearly, the test with $R = 40$, $N_1 = 383$ gives a better approximation, which verifies the discussion in "Conclusion".

With $R = 40$, Table 3 shows the errors (max and average error) with respect to different value of $N_1$ and $N_2$. The trends as functions of $N_1$ and $N_2$ are shown as plots in Figs. 9 and 10.

From Fig. 9, it can be seen that when $N_1$ individually ($N_2$ is fixed at $N_2 = 15$) is less than the minimum of 383 obtained from the sampling theorem, increasing $N_1$ will lead to smaller errors, as expected. When $N_1$ is bigger than the sampling-theorem threshold

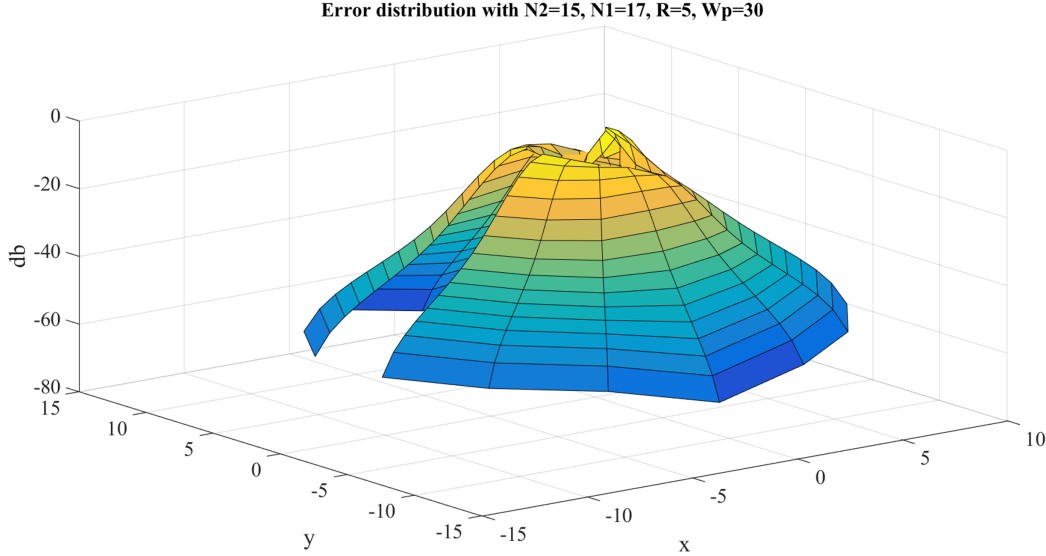

**Figure 7** Error between the sampled values of the continuous transform and the discretely calculated values for a Gaussian function with $R = 5$ and $N_1 = 17$.

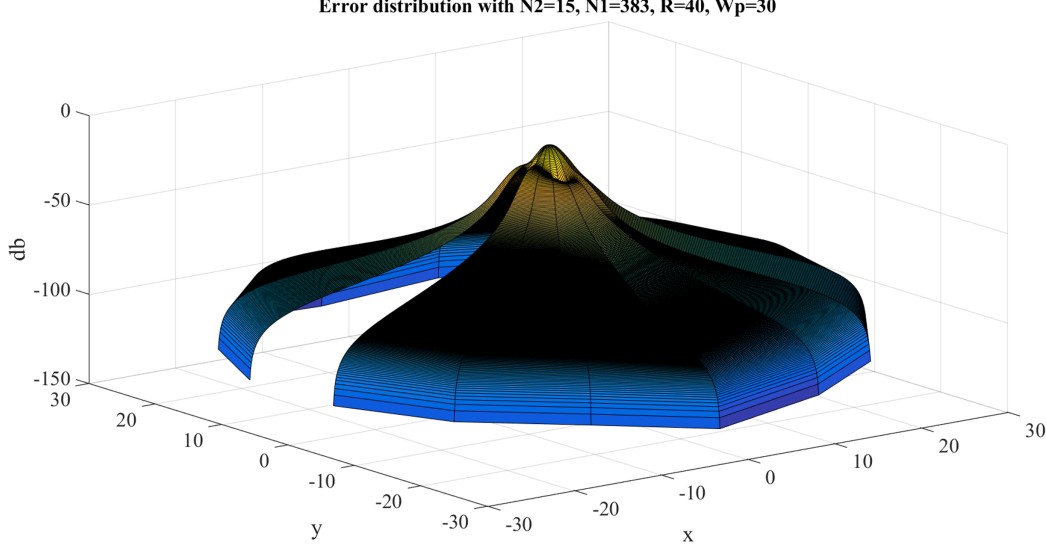

**Figure 8** Error between the sampled values of the continuous transform and the discretely calculated values for a Gaussian function with $R = 40$ and $N_1 = 383$.

of 383, increasing $N_1$ still decreases the error which verifies the discussion about sample grid coverage in "Conclusion". Increasing $N_1$ tends to increase the sample grid coverage and capture more information at the center area and thus leads to smaller errors.

From Fig. 10, increasing $N_2$ alone (i.e., without a corresponding increase in $N_1$) leads to larger errors, both Error$_{max}$ and Error$_{average}$. Although at first counterintuitive, this result is actually reasonable because the function is radially symmetric which implies that $N_2 = 1$ should be sufficient based on the sampling theorem for the angular direction.

**Table 3 Error (dB) of forward transform of Gaussian function with $R = 40$, different value of $N_1$ and $N_2$.**

| $N_2$ | $N_1$ | | | | |
|---|---|---|---|---|---|
| | 283 | 333 | 383 | 433 | 483 |
| 3 | $E_{max.} = -21.6$ | $E_{max.} = -23.0$ | $E_{max.} = -24.3$ | $E_{max.} = -25.4$ | $E_{max.} = -26.3$ |
| | $E_{avg.} = -71.3$ | $E_{avg.} = -76.9$ | $E_{avg.} = -81.8$ | $E_{avg.} = -86.0$ | $E_{avg.} = -89.8$ |
| 7 | $E_{max.} = -12.9$ | $E_{max.} = -14.4$ | $E_{max.} = -15.7$ | $E_{max.} = -16.9$ | $E_{max.} = -17.8$ |
| | $E_{avg.} = -62.6$ | $E_{avg.} = -68.3$ | $E_{avg.} = -73.2$ | $E_{avg.} = -77.5$ | $E_{avg.} = -81.4$ |
| 15 | $E_{max.} = -5.4$ | $E_{max.} = -7.0$ | $E_{max.} = -8.4$ | $E_{max.} = -9.6$ | $E_{max.} = -10.6$ |
| | $E_{avg.} = -53.1$ | $E_{avg.} = -58.9$ | $E_{avg.} = -63.8$ | $E_{avg.} = -68.1$ | $E_{avg.} = -72.0$ |
| 31 | $E_{max.} = 2.3$ | $E_{max.} = 0.5$ | $E_{max.} = -1.0$ | $E_{max.} = -2.3$ | $E_{max.} = -3.4$ |
| | $E_{avg.} = -42.0$ | $E_{avg.} = -47.6$ | $E_{avg.} = -52.5$ | $E_{avg.} = -56.9$ | $E_{avg.} = -60.7$ |
| 61 | $E_{max.} = 9.7$ | $E_{max.} = 7.9$ | $E_{max.} = 6.4$ | $E_{max.} = 5.0$ | $E_{max.} = 3.8$ |
| | $E_{avg.} = -32.5$ | $E_{avg.} = -37.5$ | $E_{avg.} = -42.0$ | $E_{avg.} = -46.1$ | $E_{avg.} = -49.8$ |

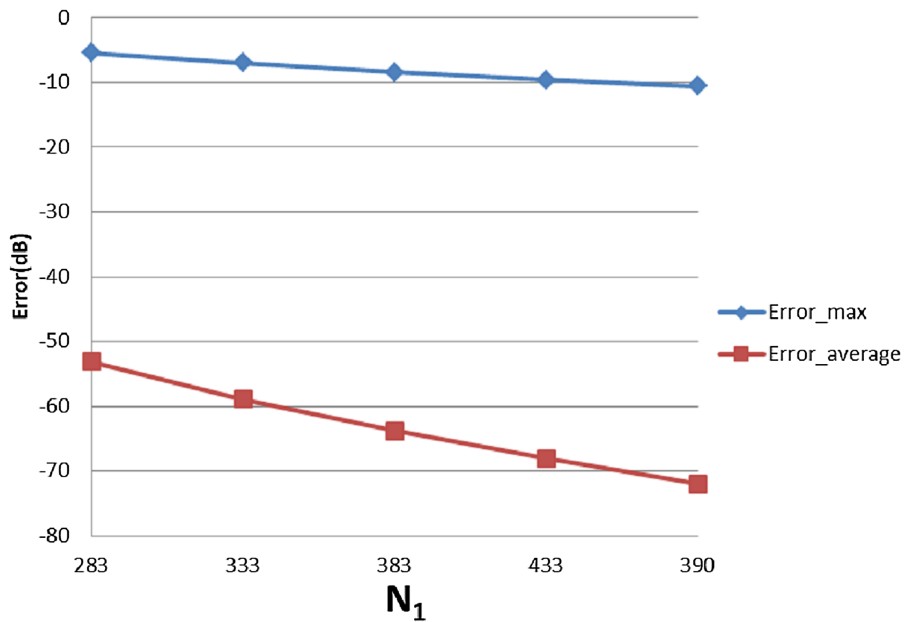

**Figure 9 Error trend between the sampled values of the continuous transform and the discretely calculated values for a Gaussian function, as a function of $N_1$.**

Therefore, increasing $N_2$ will not lead to a better approximation. Moreover, from the discussion of the sample grid coverage in "Conclusion", the sampling grid coverage in both domains gets worse when $N_2$ gets bigger because more information from the center is lost. This problem can be solved by increasing $N_1$ at the same time, but it could be computationally time consuming. Therefore, choosing $N_2$ properly is very important from the standpoint of accuracy and computational efficiency.

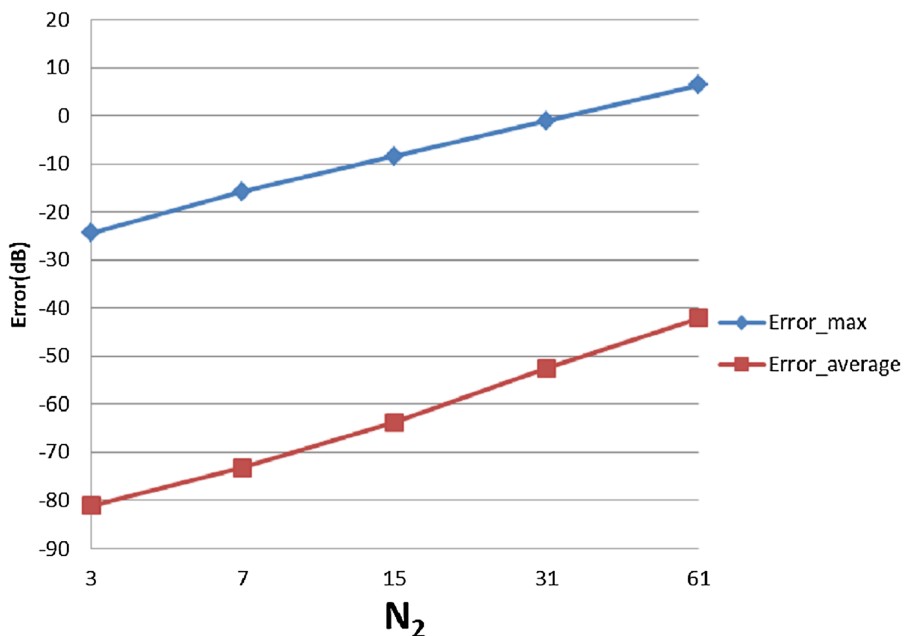

**Figure 10 Error trend between the sampled values of the continuous transform and the discretely calculated values for a Gaussian function, as a function of $N_2$.**

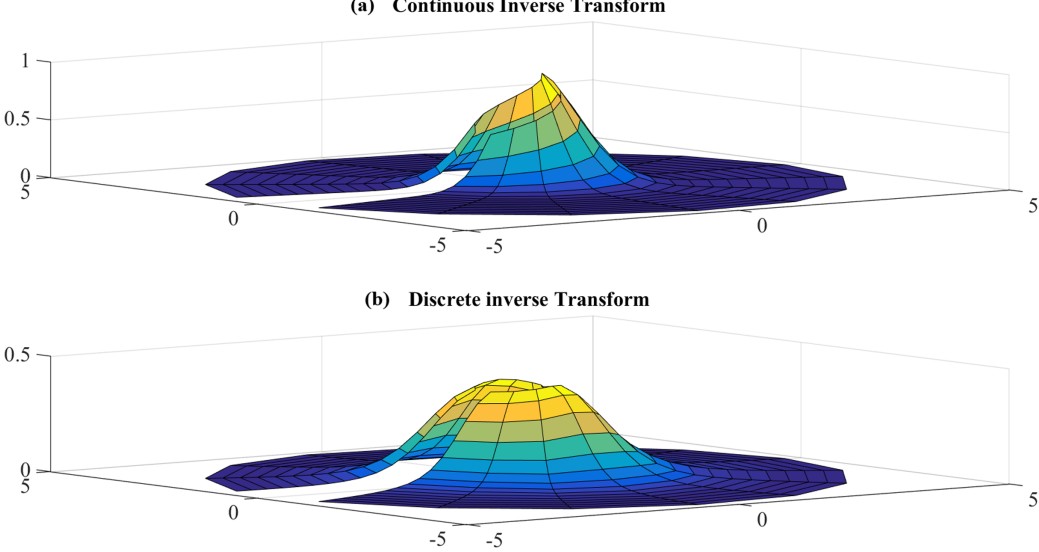

**Figure 11 (A) sampled continuous inverse transform and (B) discrete inverse transform for the Gaussian function for $R = 5$ and $N_1 = 17$.**

*Inverse transform*

Test results for the inverse transform with $R = 5$, $N_1 = 17$ are shown in Figs. 11 and 12. Figure 11 shows the sampled continuous inverse transform and discrete inverse transform and Fig. 12 shows the error between the sampled continuous and discretely calculated values.

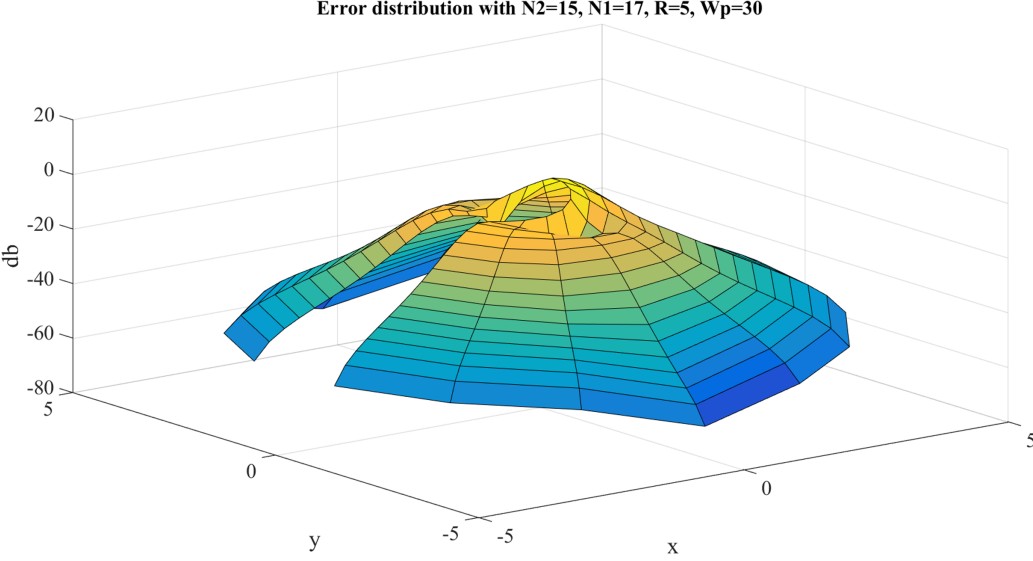

**Figure 12 Error between the sampled continuous inverse transform and discrete inverse transform for the Gaussian function for $R = 5$ and $N_1 = 17$.**

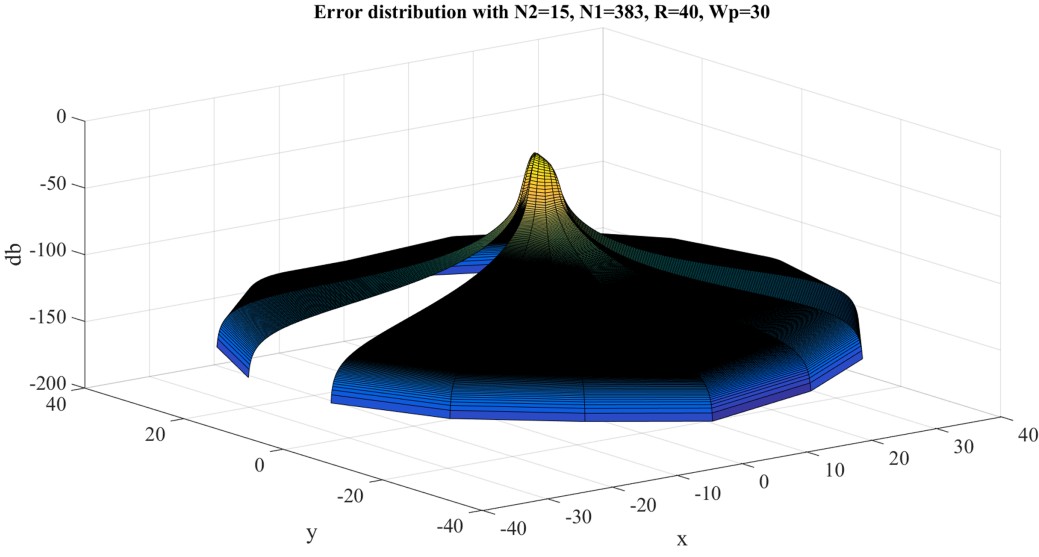

**Figure 13 Error between the sampled continuous inverse transform and discrete inverse transform for the Gaussian function for $R = 40$ and $N_1 = 383$.**

Similar to the case for the forward transform, the error gets larger at the center, which is as expected because the sampling grid shows that the sampling points never attain the center. The maximum value of the error is $E_{\text{max}} = 3.1954$ dB and this occurs at the center. The average of the error is $E_{\text{avg.}} = -25.7799$ dB.

Error test results for the inverse transform with $R = 40$, $N_1 = 383$ are shown in Fig. 13. In this case, the maximum value of the error is $E_{\text{max}} = -12.2602$ dB and this occurs at the center. The average of the error is $E_{\text{avg.}} = -98.0316$ dB. Table 4 shows the errors with respect to different value of $N_1$ and $N_2$, from which Figs. 14 and 15 demonstrate the trend.

**Table 4  Error (dB) of inverse transform of Gaussian function with $R = 40$, different value of $N_1$ and $N_2$.**

| $N_2$ | $N_1$ | | | | |
|---|---|---|---|---|---|
| | 283 | 333 | 383 | 433 | 483 |
| 3 | $E_{max.} = -25.9$ | $E_{max.} = -27.5$ | $E_{max.} = -28.9$ | $E_{max.} = -30.2$ | $E_{max.} = -31.3$ |
| | $E_{avg.} = -115.3$ | $E_{avg.} = -115.4$ | $E_{avg.} = -115.4$ | $E_{avg.} = -115.5$ | $E_{avg.} = -115.5$ |
| 7 | $E_{max.} = -16.5$ | $E_{max.} = -18.1$ | $E_{max.} = -19.4$ | $E_{max.} = -20.5$ | $E_{max.} = -21.6$ |
| | $E_{avg.} = -107.0$ | $E_{avg.} = -107.1$ | $E_{avg.} = -107.2$ | $E_{avg.} = -107.2$ | $E_{avg.} = -107.2$ |
| 15 | $E_{max.} = -9.7$ | $E_{max.} = -11.0$ | $E_{max.} = -12.3$ | $E_{max.} = -13.4$ | $E_{max.} = -14.4$ |
| | $E_{avg.} = -97.9$ | $E_{avg.} = -98.0$ | $E_{avg.} = -98.0$ | $E_{avg.} = -98.1$ | $E_{avg.} = -98.1$ |
| 34 | $E_{max.} = -4.4$ | $E_{max.} = -5.5$ | $E_{max.} = -6.5$ | $E_{max.} = -7.5$ | $E_{max.} = -8.3$ |
| | $E_{avg.} = -86.9$ | $E_{avg.} = -86.9$ | $E_{avg.} = -87.0$ | $E_{avg.} = -87.0$ | $E_{avg.} = -87.0$ |
| 61 | $E_{max.} = -1.1$ | $E_{max.} = -1.7$ | $E_{max.} = -2.4$ | $E_{max.} = -3.0$ | $E_{max.} = -3.7$ |
| | $E_{avg.} = -75.6$ | $E_{avg.} = -75.6$ | $E_{avg.} = -75.6$ | $E_{avg.} = -75.6$ | $E_{avg.} = -75.7$ |

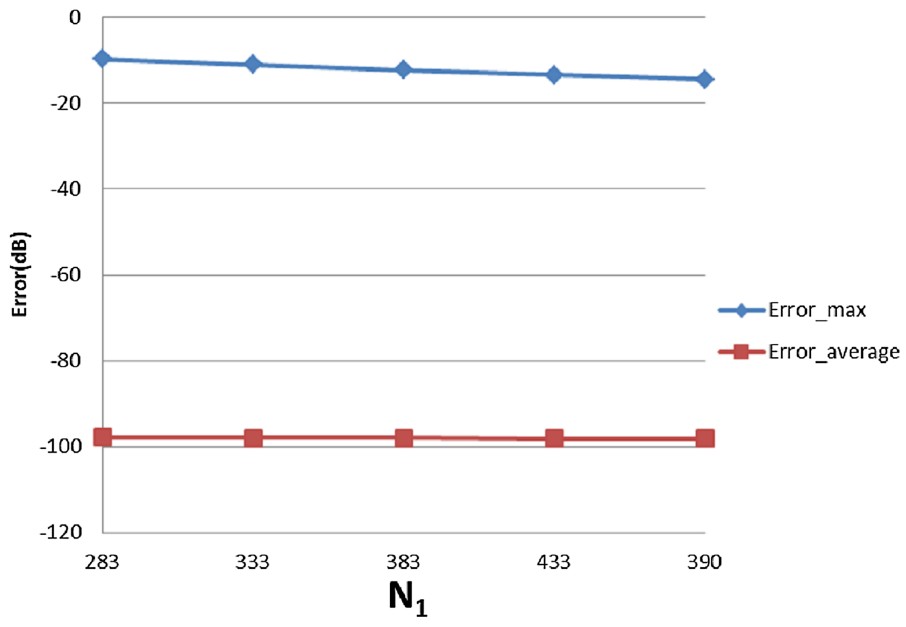

**Figure 14 Error trend between the sampled values of the continuous inverse transform and the discretely calculated values for a Gaussian function, as a function of $N_1$.**

From Fig. 15 it can be observed that increasing $N_1$ tends to improve the result but not significantly. This could be explained by the discussion for $R = 40$, $N_1 = 383$ that with fixed $R$ and $W_\rho$, increasing $N_1$ will not allow the sampling grid in the frequency domain to get any closer to the origin to capture more information. From Fig. 15, increasing $N_2$ (with fixed $N_1 = 383$) leads to a worse approximation which verifies the discussion for $R = 40$, $N_1 = 383$.

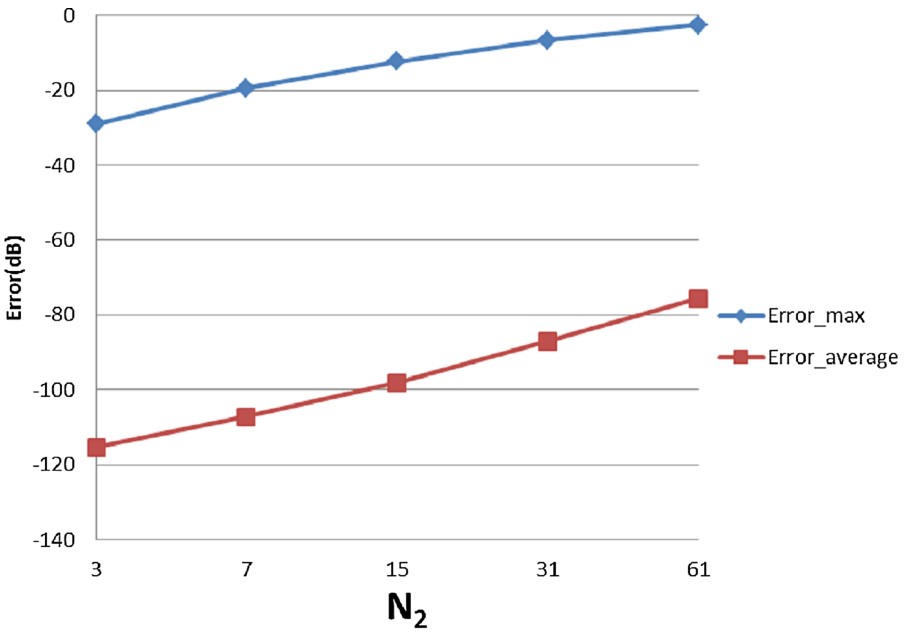

**Figure 15 Error trend between the sampled values of the continuous inverse transform and the discretely calculated values for a Gaussian function, as a function of $N_2$.**

Performing sequential 2D-DFT and 2D-IDFT results in $\varepsilon = 4.1656 \times e^{-17}$ where $\varepsilon$ is calculated with Eq. (51). Therefore, performing sequential forward and inverse transforms does not add much error.

### Four-term sinusoid & Sinc function

The second function chosen for evaluation is given by

$$f(r, \theta) = \frac{\sin(ar)}{ar}[3\sin(\theta) + \sin(3\theta) + 4\cos(10\theta) + 12\sin(15\theta)] \tag{54}$$

which is a sinc function in the radial direction and a four-term sinusoid in the angular direction. The graphs for the original function and the magnitude of its continuous 2D-FT with $a = 5$ are shown in Fig. 16. From Fig. 16, the function can be considered as a band-limited function. Therefore Eqs. (16) and (17) were used to implement the forward and inverse transform.

The continuous 2D-FT can be calculated from *Baddour (2011)*

$$F(\rho, \psi) = \sum_{n=-\infty}^{\infty} 2\pi i^{-n} e^{in\psi} \int_0^\infty f_n(r) J_n(\rho r) r \, dr \tag{55}$$

where $f_n(r)$ is the Fourier series of $f(r, \theta)$ and can be written as

$$f_n(r) = \frac{1}{2\pi} \int_{-\pi}^{\pi} f(r, \theta) e^{-in\theta} d\theta \tag{56}$$

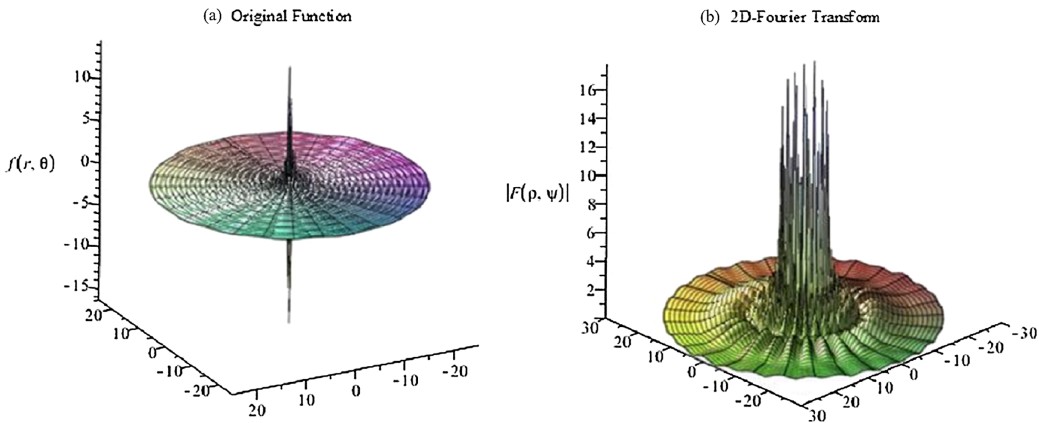

**Figure 16 Plots of the (A) original function (four-term sinusoid and sinc) and (B) the magnitude of its continuous forward 2D Fourier transform with $a = 5$.**

From the sampling theorem for the angular direction, the highest angular frequency in Eq. (54) results in $N_2$ being at least 31 required to reconstruct the signal. Therefore, at least 31 terms are required to calculate the continuous 2D-FT, which can be written as

$$F(\rho,\psi)=$$
$$\begin{cases} \dfrac{8\pi\cos(10\psi)\rho^{10}}{a\sqrt{a^2-\rho^2}\left(a+\sqrt{a^2-\rho^2}\right)^{10}}, \ \rho<a \\[2ex] -\dfrac{6\pi i\sin(\psi)}{a\rho\sqrt{\rho^2+a^2}}+\dfrac{2\pi i\sin\left(3\arcsin\left(\dfrac{a}{\rho}\right)\right)\sin(3\psi)}{\sqrt{\rho^2+a^2}}-\dfrac{8\pi\sin\left(10\arcsin\left(\dfrac{a}{\rho}\right)\right)\cos(10\psi)}{\sqrt{\rho^2+a^2}} \\[2ex] +\dfrac{24\pi i\sin\left(15\arcsin\left(\dfrac{a}{\rho}\right)\right)\sin(15\psi)}{\sqrt{\rho^2+a^2}}, \ \rho>a \end{cases} \quad (57)$$

In the angular direction, the highest frequency term in the function in the space domain is $12\sin(15\theta)$. From the sampling theorem, the sampling frequency should be at least twice that of the highest frequency present in the signal. Thus, $N_2 = 41$ is chosen in order to go a little past the minimum requirement of 31. In the radial direction, from the graphs of the original function and its 2D-FT, it can be assumed that $f(r,\theta)$ is space-limited at $R = 15$ and band-limited at $W_\rho = 30$. However, since most of the energy in the space domain is located at the origin, a relatively large band limit should be chosen based on the discussion in "Conclusion". Therefore, $W_\rho = 90$, $N_1 = 430$ are chosen.

*Forward transform*
The error results for the forward 2D-DFT of Four-term sinusoid & Sinc function with $W_\rho = 90$, $N_1 = 430$ are shown in Fig. 17. The discrete transform does not approximate the continuous transform very well. This is expected because the function in the frequency domain is discontinuous and the sampling points close to the discontinuity will result in a very large error. The maximum value of the error is $\mathrm{Error}_{max} = 10.6535$ dB

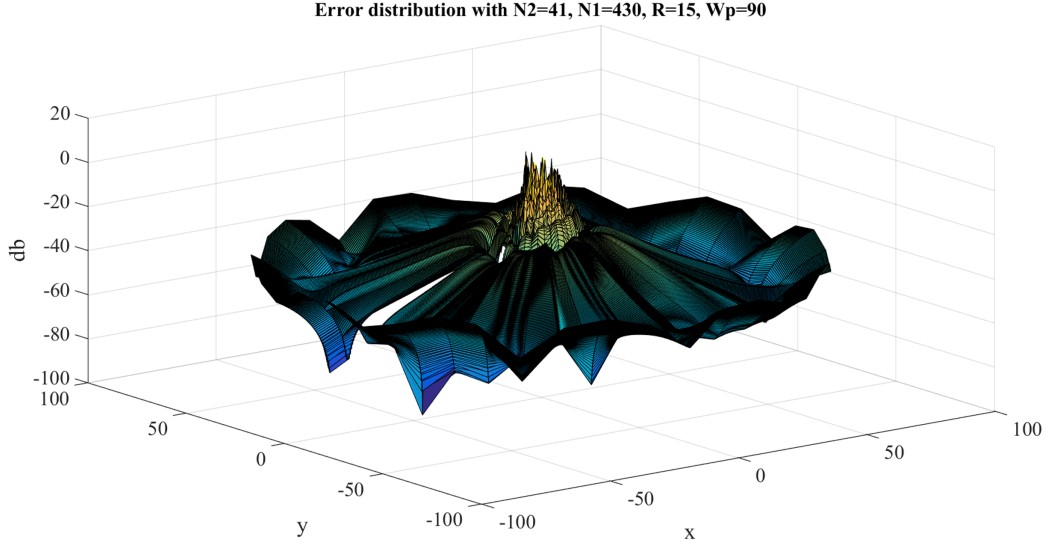

**Error distribution with N2=41, N1=430, R=15, Wp=90**

**Figure 17 Error results for the forward 2D Fourier transform of the Four-term sinusoid & Sinc function for $W_p = 90$ and $N_1 = 430$.**

**Table 5 Error (dB) of the forward transform of 'four-term sinusoid & Sinc' function with different value of $N_1$ and $N_2$ of forward transform.**

| $N_2$ | $N_1$ | | | | |
|---|---|---|---|---|---|
| | 330 | 380 | 430 | 480 | 530 |
| 11 | $E_{max.}$ = 4.6 | $E_{max.}$ = 7.1 | $E_{max.}$ = 3.4 | $E_{max.}$ = 9.0 | $E_{max.}$ = 2.8 |
| | $E_{avg.}$ = −33.6 | $E_{avg.}$ = −33.4 | $E_{avg.}$ = −33.5 | $E_{avg.}$ = −35.1 | $E_{avg.}$ = −35.5 |
| 21 | $E_{max.}$ = 6.7 | $E_{max.}$ = 10.5 | $E_{max.}$ = 3.2 | $E_{max.}$ = 6.9 | $E_{max.}$ = 3.6 |
| | $E_{avg.}$ = −33.9 | $E_{avg.}$ = −34.6 | $E_{avg.}$ = −37.2 | $E_{avg.}$ = −38.0 | $E_{avg.}$ = −38.1 |
| 41 | $E_{max.}$ = 8.5 | $E_{max.}$ = 35.1 | $E_{max.}$ = 10.7 | $E_{max.}$ = 14.6 | $E_{max.}$ = 11.1 |
| | $E_{avg.}$ = −38.7 | $E_{avg.}$ = −38.9 | $E_{avg.}$ = −38.8 | $E_{avg.}$ = −39.8 | $E_{avg.}$ = −41.3 |
| 81 | $E_{max.}$ = 9.7 | $E_{max.}$ = 32.7 | $E_{max.}$ = 14.8 | $E_{max.}$ = 22.6 | $E_{max.}$ = 14.5 |
| | $E_{avg.}$ = −34.3 | $E_{avg.}$ = 35.5 | $E_{avg.}$ = −36.2 | $E_{avg.}$ = −37.3 | $E_{avg.}$ = −37.5 |
| 161 | $E_{max.}$ = 19.9 | $E_{max.}$ = 30.2 | $E_{max.}$ = 22.5 | $E_{max.}$ = 22.5 | $E_{max.}$ = 16.1 |
| | $E_{avg.}$ = −29.4 | $E_{avg.}$ = −30.7 | $E_{avg.}$ = −31.1 | $E_{avg.}$ = −32.2 | $E_{avg.}$ = −32.8 |

and this occurs where the discontinuities are located. The average of the error is $\text{Error}_{average} = -38.7831$ dB.

With $W_\rho = 90$, $N_1 = 430$, Table 5 shows the errors with respect to different value of $N_1$ and $N_2$, from which Figs. 18 and 19 show the trend. From Fig. 18, increasing $N_1$ alone tends improve the average error. The maximum error does not change with $N_1$, which is reasonable because of the discontinuity of the function in the frequency domain.

From Fig. 19, increasing $N_2$ leads to $\text{Error}_{max}$ and $\text{Error}_{average}$ first improving and then worsening. This is reasonable because when $N_2$ is less than the minimum requirement of 31 from sampling theorem, the test result is actually affected by both sampling point density (from the sampling theorem) and grid coverage (discussed in "Conclusion").

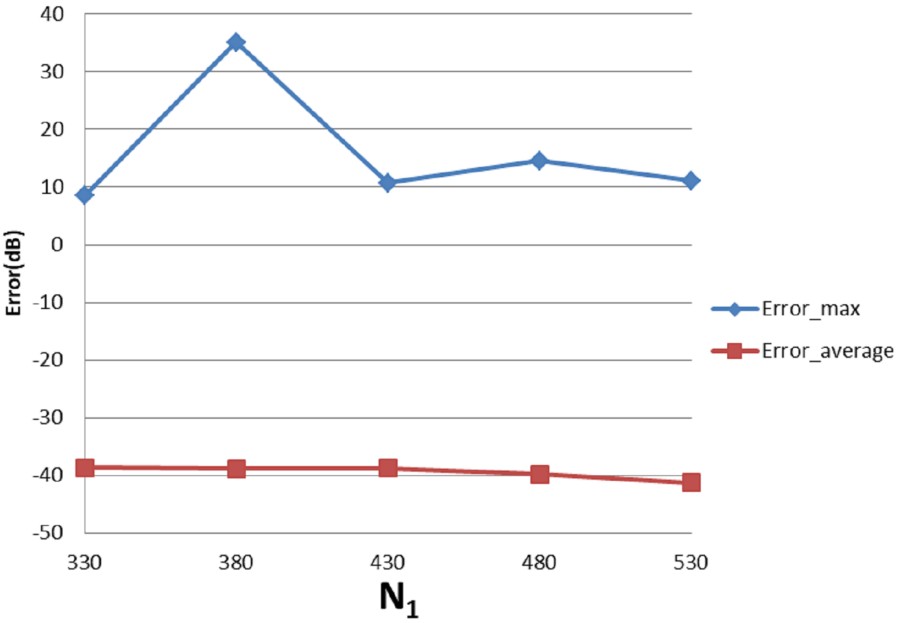

**Figure 18 Error trend between the sampled values of the continuous forward transform and the discretely calculated values for a four-term sinusoid and sinc as a function of $N_1$.**

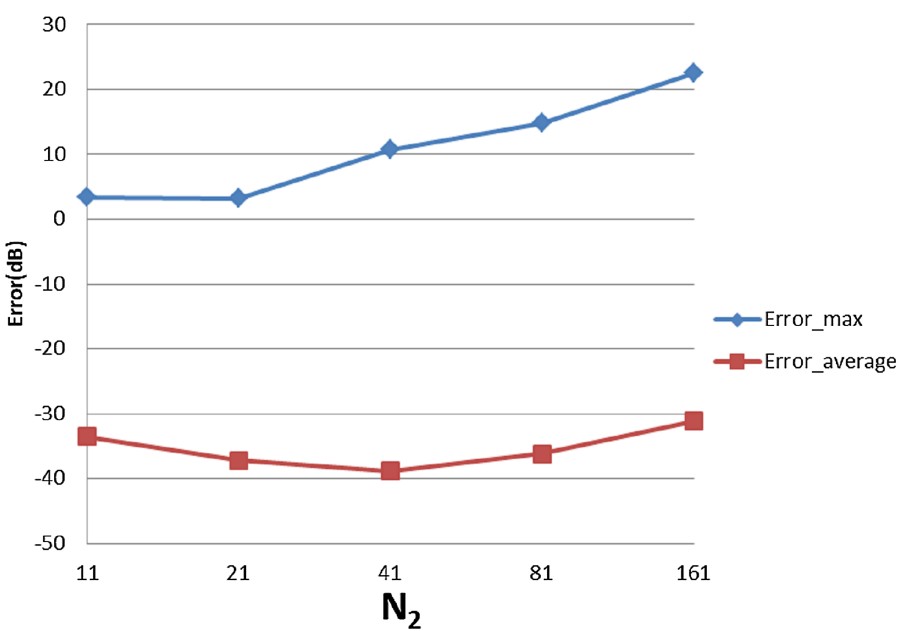

**Figure 19 Error trend between the sampled values of the continuous forward transform and the discretely calculated values for a four-term sinusoid and sinc as a function of $N_2$.**

Increasing $N_2$ should give better results from the point of view of the sampling theorem but worse grid coverage. The result from the combined effects is dependent on the function properties. In the specific case of this function, when $N_2$ is bigger than 31, thereby

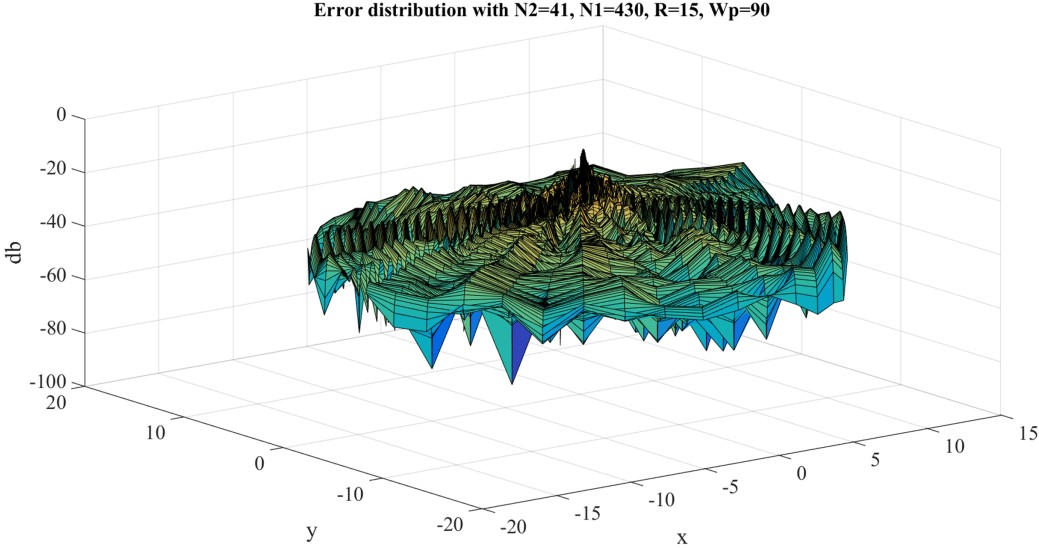

**Error distribution with N2=41, N1=430, R=15, Wp=90**

**Figure 20 Error results for the 2D inverse discrete Fourier transform of the four-term sinusoid and sinc function for $W_P = 90$ and $N_1 = 430$.**

**Table 6 Error (dB) of inverse transform of 'four-term sinusoid & Sinc' function with different value of $N_1$ and $N_2$.**

| $N_2$ | $N_1$ | | | | |
|---|---|---|---|---|---|
| | 330 | 380 | 430 | 480 | 530 |
| 11 | $E_{max.} = 0.1$ | $E_{max.} = 0.1$ | $E_{max.} = 0.1$ | $E_{max.} = 0.1$ | $E_{max.} = 0.1$ |
| | $E_{avg.} = -43.7$ | $E_{avg.} = -43.7$ | $E_{avg.} = -46.6$ | $E_{avg.} = -45.6$ | $E_{avg.} = -48.1$ |
| 21 | $E_{max.} = 0.7$ | $E_{max.} = 0.7$ | $E_{max.} = 0.6$ | $E_{max.} = 0.6$ | $E_{max.} = 0.7$ |
| | $E_{avg.} = -38.3$ | $E_{avg.} = -38.0$ | $E_{avg.} = -40.4$ | $E_{avg.} = -40.6$ | $E_{avg.} = -42.2$ |
| 41 | $E_{max.} = -9.0$ | $E_{max.} = -8.5$ | $E_{max.} = -8.7$ | $E_{max.} = -8.8$ | $E_{max.} = -8.6$ |
| | $E_{avg.} = -35.9$ | $E_{avg.} = -24.7$ | $E_{avg.} = -37.8$ | $E_{avg.} = -38.2$ | $E_{avg.} = -39.0$ |
| 81 | $E_{max.} = -4.5$ | $E_{max.} = -4.7$ | $E_{max.} = -4.5$ | $E_{max.} = -4.6$ | $E_{max.} = -4.5$ |
| | $E_{avg.} = -35.7$ | $E_{avg.} = -26.5$ | $E_{avg.} = -37.5$ | $E_{avg.} = -36.2$ | $E_{avg.} = -39.0$ |
| 161 | $E_{max.} = 0.8$ | $E_{max.} = 0.7$ | $E_{max.} = 0.7$ | $E_{max.} = 0.7$ | $E_{max.} = 0.7$ |
| | $E_{avg.} = -35.6$ | $E_{avg.} = -32.5$ | $E_{avg.} = -36.6$ | $E_{avg.} = -37.2$ | $E_{avg.} = -39.2$ |

implying that the angular sampling theorem has been satisfied—the results get worse with increasing $N_2$.

*Inverse transform*

The error results for the 2D-IDFT of Four-term sinusoid & Sinc function with $W_\rho = 90$, $N_1 = 430$ are shown in Fig. 20. The maximum value of the error is Error$_{max}$ = $-8.6734$ dB. The average of the error is Error$_{average}$ = $-37.8119$ dB. With $W_\rho = 90$, $N_1 = 430$, Table 6 shows the errors with respect to different value of $N_1$ and $N_2$, from which Figs. 21 and 22 show the trend.

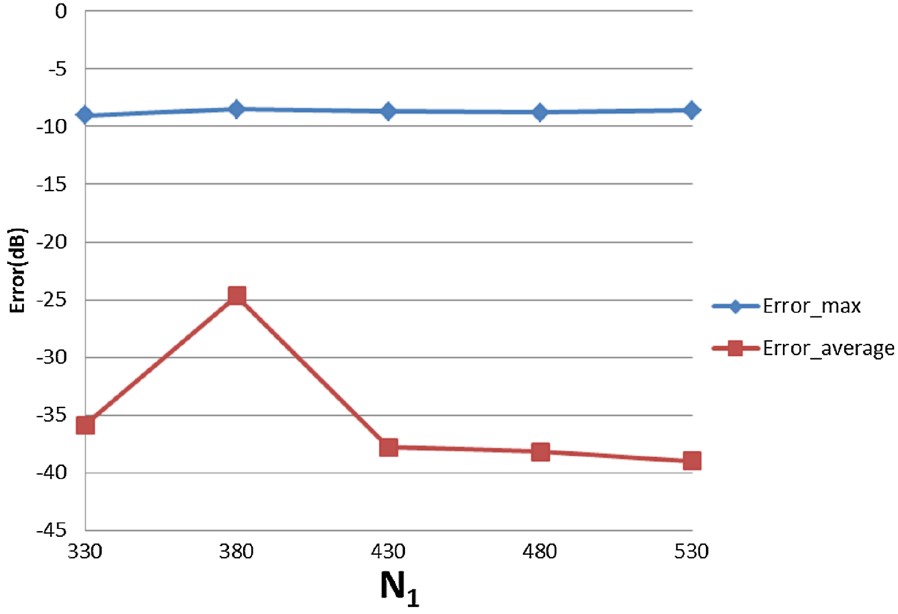

**Figure 21 Error trend between the sampled values of the continuous inverse transform and the discretely calculated values for a four-term sinusoid and sinc function, as a function of $N_1$.**

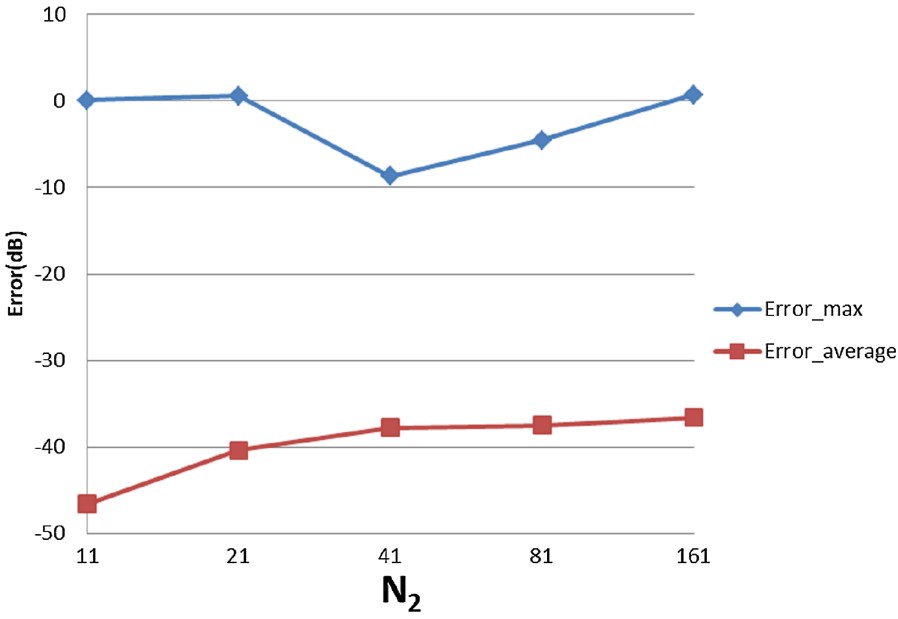

**Figure 22 Error trend between the sampled values of the continuous inverse transform and the discretely calculated values for a four-term sinusoid and sinc function, as a function of $N_2$.**

From Fig. 21, it can be observed that the increasing $N_1$ alone improves the average error, as was expected. However, $N_1 = 380$ gives an apparently worse average error than the other points. This could be caused by the discontinuity of the function in the frequency

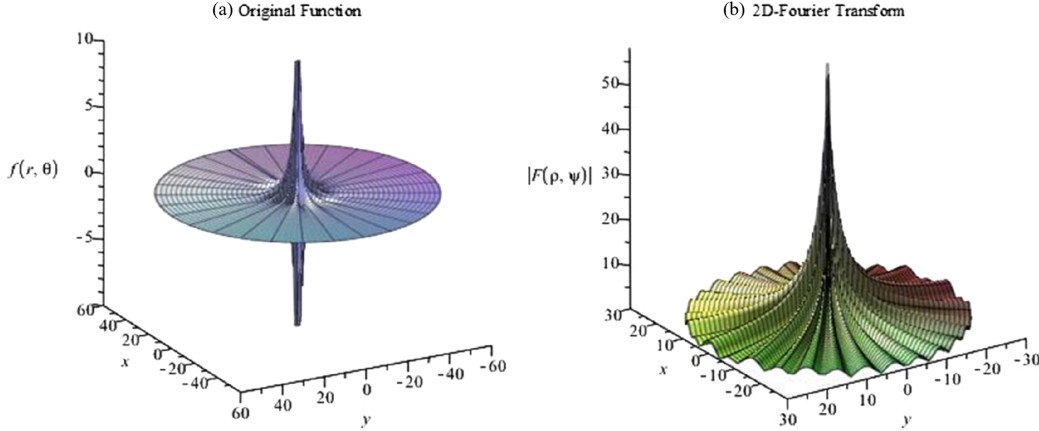

**Figure 23 Plots for (A) the original function and (B) the magnitude of its continuous 2D discrete Fourier transform with $a = 0.1$ for a four-term sinusoid and modified exponential function.**

domain. Changing to $N_1 = 381$, the average error becomes $-37.0$ dB which proves that the large error is caused by the discontinuity.

From Fig. 22, increasing $N_2$ does not lead to worse results, which is different from previous cases. However, from Fig. 16 it can be seen that the function in the frequency domain does not have much information in the center area. So, even though increasing $N_2$ causes a larger hole in the center as discussed in "Conclusion", it does not lead to losing much energy which explains why Fig. 22 shows a different trend from the previous cases.

Performing 2D-DFT and 2D-IDFT sequentially results in $\varepsilon = 1.3117 \times e^{-12}$ where $\varepsilon$ is calculated by Eq. (51).

### Four-term sinusoid and modified exponential

For the next test function, the function is given by

$$f(r, \theta) = \frac{e^{-ar}}{r}[3\sin(\theta) + \sin(3\theta) + 4\cos(10\theta) + 12\sin(15\theta)] \tag{58}$$

Its continuous 2D-FT can be calculated as

$$
F(\rho, \psi) = -6\pi i \sin(\psi)\frac{\sqrt{\rho^2 + a^2} - a}{\rho\sqrt{\rho^2 + a^2}} + 2\pi i \sin(3\psi)\frac{\left(\sqrt{\rho^2 + a^2} - a\right)^3}{\rho^3\sqrt{\rho^2 + a^2}}
$$
$$
- 8\pi\cos(10\psi)\frac{\left(\sqrt{\rho^2 + a^2} - a\right)^{10}}{\rho^{10}\sqrt{\rho^2 + a^2}} + 24\pi i \sin(15\psi)\frac{\left(\sqrt{\rho^2 + a^2} - a\right)^{15}}{\rho^{15}\sqrt{\rho^2 + a^2}} \tag{59}
$$

The graphs for the original function and the magnitude of its continuous 2D-FT with $a = 0.1$ are shown in Fig. 23. From Fig. 23, it can be observed that the function has a spike in both domains, which is a more difficult scenario based on the discussion in "Conclusion". In this case, the function can be assumed as space-limited or band-limited. This function will be tested as being space-limited.

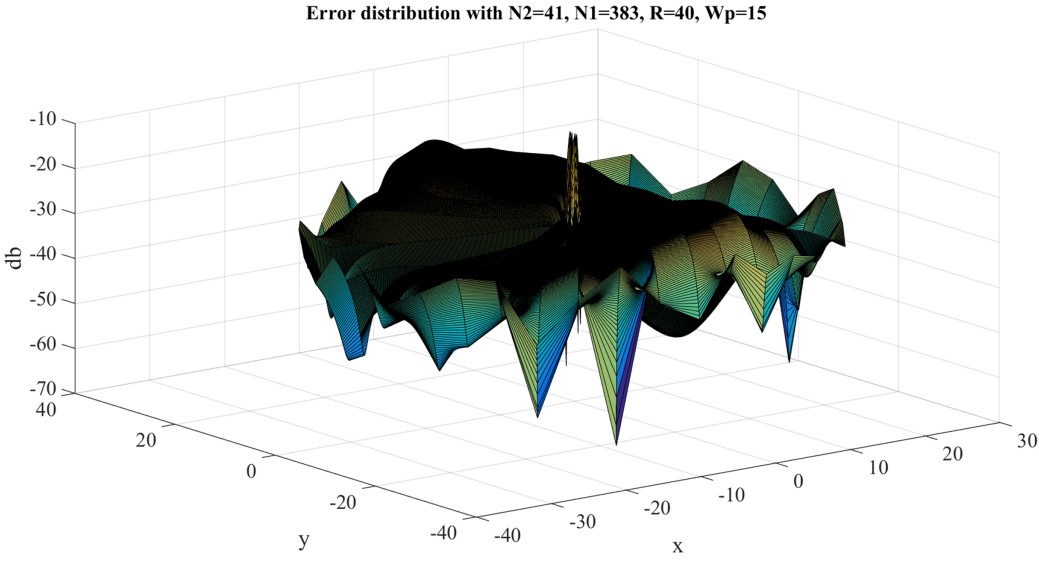

**Figure 24 Error between the sampled values of the continuous forward transform and the discretely calculated values for the four-term sinusoid and modified exponential function with $R = 40$, $W_p = 30$ and $N_1 = 383$.**

From graph of the original function and its 2D-DFT, it can be assumed that $f(r, \theta)$ is effectively space-limited with $R = 20$, and $F(\rho, \psi)$ is effectively band-limited with $W_\rho = 15$, which gives $j_{0N_1} \approx 300$. This results in $N_1 = 96$. However, since the function explodes at the center area in both domains, relatively large values of $R$ and $W_\rho$ should give a better approximation. Therefore, another case with $R = 40$, $W_\rho = 30$ is tested. In this case, $N_1 = 383$ is chosen.

In the angular direction, the highest frequency term is $12 \sin(15\theta)$. From the sampling theorem, the sampling frequency should be at least twice of the highest frequency of signal. Thus, $N_2 = 41$ is chosen.

*Forward transform*

Here, the function is tested as a space limited function and Eqs. (14) and (15) are used to proceed with the forward and inverse transform in sequence. The error results with $R = 40$, $W_\rho = 30$, $N_1 = 383$ are shown in Fig. 24. The maximum value of the error is $\text{Error}_{\text{max}} = -10.1535$ dB and this occurs at the center area. The average of the error is $\text{Error}_{\text{average}} = -32.7619$ dB. This demonstrates that the discrete function approximates the sampled values of the continuous function quite well.

*Inverse transform*

The error results with $R = 40$, $W_\rho = 30$, $N_1 = 383$ are shown in Fig. 25.

The maximum value of the error is $\text{Error}_{\text{max}} = 0.5579$ dB and this occurs at the center. The average of the error isError$_{\text{average}} = -68.7317$ dB.

Performing 2D-DFT and 2D-IDFT results in $\varepsilon = 1.421 \times e^{-12}$, where $\varepsilon$ is calculated by Eq. (51).

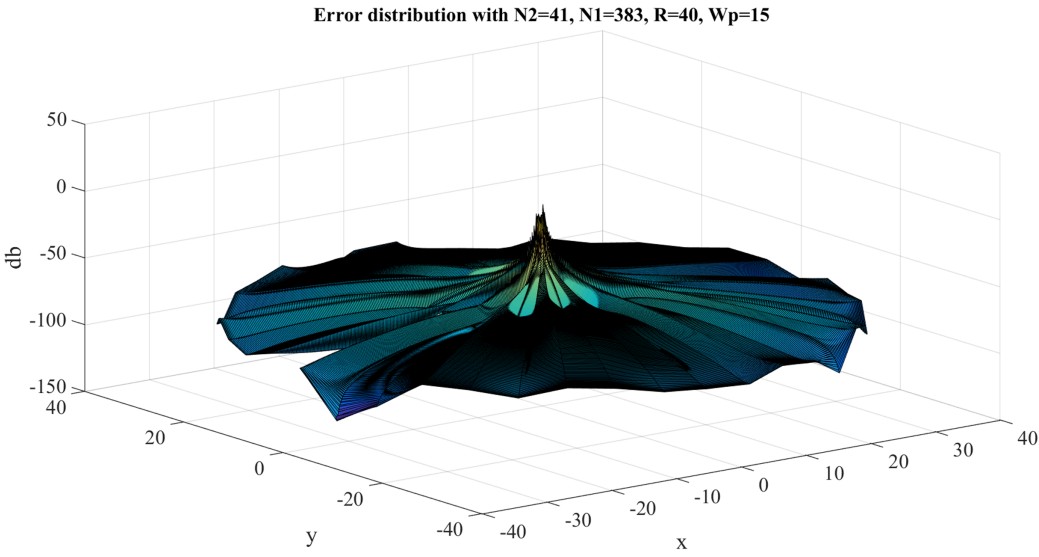

**Figure 25 Error between the sampled values of the continuous inverse transform and the discretely calculated values for the four-term sinusoid and modified exponential function with $R = 40$, $W_p = 30$ and $N_1 = 383$.**

**Table 7 Computing time of three cases: Case1: Run the transform as matrixes in matrix without pre-calculating the Bessel zeros; Case2: Run the transform as DFT, DHT and IDFT in sequence without pre-calculating the Bessel zeros; Case3: Run the transform as DFT, DHT.**

| Test cases | Total computing time (s) |
| --- | --- |
| Case 1 | 3,346.5 |
| Case 2 | 321.1 |
| Case 3 | 14.3 |

It can be observed that even for functions with the worst properties, the proposed transform can still be used to approximate the continuous FT with fairly small errors, as long as the function is sampled properly.

## SUMMARY AND CONCLUSION

### Accuracy and precision of the transform

The proposed discrete 2D-FT in polar coordinates demonstrates an acceptable accuracy in providing discrete estimates to the continuous FT in polar coordinates. In *Baddour & Chouinard (2015)*, *Guizar-Sicairos & Gutiérrez-Vega (2004)* and *Higgins & Munson (1987)*, the one dimensional Hankel transform of a sinc function showed similar dynamic error, which could be used as a comparative measure. Since the DHT is one step of the proposed discrete 2D-FT, and the definition of the Hankel transform is based on *Abbas, Sun & Foroosh (2017)*, a similar dynamic error should be expected.

The test cases showed that the transform introduced very small errors ($\varepsilon = 1.4004 \times e^{-12}$ for worst case) by performing a forward transform and an inverse

transform sequentially, which demonstrates that the discrete transform shows good precision.

### Guidelines of choosing sample size

As discussed in "Conclusion" and proved by test cases, the sample size $N_1$ (sample size in the radial direction) and $N_2$ (sample size in the angular direction) do not have to be of the same order. For functions with different properties, sample size in the different directions could be very different. To approximate the continuous FT properly, sample size should be chosen based on the discussion in "Conclusion".

### Interpretation of the transform

By interpreting the transform as a 1DDFT, 1D DHT and 1D IDFT, the computing time of the transform is improved to a useful level since the FFT can be used to compute the DFT.

## APPENDIX: IMPROVING THE COMPUTING TIME OF THE TRANSFORM

One of the advantages of the traditional FT is that the computing speed is fast by using the now well-established *fft* algorithm. To reduce the computing time of the 2D DFT in polar coordinates, the following steps are recommended:

1. Interpreting the transform as three sequential operations (DFT, DHT, IDFT) instead of a single four-dimensional matrix.

2. Pre-calculating and saving the Bessel zeros.

### Reducing computing time by interpreting the transform as three operations in sequence

As explained above, the essence of the transform is that the matrix $f_{pk}$ is transformed into the matrix $F_{ql}$. The intuitive way to interpret the transform kernel is to think of it as a four-dimensional matrix or matrices in a matrix. However, interpreting the transform as a 1D-DFT of each column, a 1D-DHT of each row and then a 1D-IDFT of each column makes it possible to use the Matlab built in functions *fft* and *ifft*, which significantly reduced the computational time.

### Reduce computing time by pre-calculating the Bessel Zeros

After defining the transform as three operations in sequence and using the matrix for the DHT defined in *Lozier (2003)*, it was found that a lot of computational time was used to calculate the Bessel zeros for every different test case, even though the Bessel zeros are the same in every case. Pre-calculating the Bessel zeros and storing the results for large numbers of $N_1$ and $N_2$ saves a lot of time.

Table 7 shows the computing time of a forward transform on the same computer (Processor: Intel(R) Core(TM) i7-4710HQ CPU, RAM:12GB) for three cases:

1. Evaluate the transform as matrices in a matrix without pre-calculating the Bessel zeros.

2. Evaluate the transform as a DFT, DHT and IDFT in sequence without pre-calculating the Bessel zeros.

3. Evaluate the transform as a DFT, DHT and IDFT in sequence with pre-calculating the Bessel zeros.

The Gaussian function was used as the test function therefore $N_1 = 383$ and $N_2 = 15$. From Table 7, it can be clearly observed that the computing time by running the transform as matrices in a matrix costs 3,346.5 s, which is not acceptable for the transform to be useful. Testing the transform as three operations turns 3,346.5 s into 321.1 s. This is much better. Finally, pre-calculating the Bessel Zeros makes the transform much faster and applicable.

### Funding
This work was financially supported by the Natural Sciences and Engineering Research Council of Canada, grant number RGPIN-2016-04190. The funders had no role in study design, data collection and analysis, decision to publish, or preparation of the manuscript.

### Grant Disclosures
The following grant information was disclosed by the authors:
Natural Sciences and Engineering Research Council of Canada: RGPIN-2016-04190.

### Competing Interests
The authors declare that they have no competing interests.

### Author Contributions
- Xueyang Yao performed the experiments, analyzed the data, performed the computation work, prepared figures and/or tables, authored or reviewed drafts of the paper, and approved the final draft.
- Natalie Baddour conceived and designed the experiments, analyzed the data, authored or reviewed drafts of the paper, and approved the final draft.

### Data Availability
Sample Matlab code is available as a Supplemental File.

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
