# Peer review of "Discrete two dimensional Fourier transform in polar coordinates part II: numerical computation and approximation of the continuous transform"

_PeerJ Computer Science, doi:10.7717/peerj-cs.257_

## Round 0.1 · original submission · Major Revisions

We have received mixed reviews. One accept, one reject, one major revision. One reviewer's report is very specific and all reviewers raised some very good points. Please address these points carefully.

·

Basic reporting

This is a clear and well written manuscript. It has sufficient references. It is self contained with relevant results.

Experimental design

The present paper the authors discuss the computations aspects ıof the two-dimensional Dicrete Fourier Transform (2D DFT) in polar coordinates. They demonstrate how the decomposition of the 2D DFT as a DFT, Discrete Hankel Transform (DHT) and inverse DFT sequence can be exploited for efficient code. It is also demonstrated that the new 2D DFT can be used to approximate the continuous forward and inverse Fourier transform in polar coordinates.

Validity of the findings

Nothing to add.

Additional comments

I recommend the publication of this article.

Reviewer 2 ·

Basic reporting

This paper is well organized.

Experimental design

I do not understand what is actually the difference between the "sampled continuous forward transform" and the "discrete forward transform". There should be no difference.

Validity of the findings

I doubt that these findings can withstand a rigorous check from the mathematical point of view.

Additional comments

Looking at this text and the ones previously published I feel like a beginner. Many, many things are not understood. First of all, I do not understand the motivation for the "Polar 2D DFT" and the way it will outperform the standard DFT. Second, the DFT arises very naturally because functions are truncated in time and frequency which means they are periodized in time and frequency. It leads naturally to the DFT which treats functions as if they were periodic in both domains and because periodization in one domain means discretization in the other domain we actually obtain functions which are discrete in time and frequency, simultaneously. Third, the treatment of the Dirac impulse is not done in the generalized functions sense and I saw many mistakes regarding the Dirac impulse. I do not doubt that these results might be usable in some way but I believe they are mathematically not rigorously deduced. Due to the many indices and technical details, I also doubt that these results will indeed be used by researchers who are willing to understand and use it.

·

Basic reporting

see full report attached

Experimental design

see full report attached

Validity of the findings

see full report attached

Additional comments

see full report attached

---

## Round 0.2 · Minor Revisions

The reviewer believes this paper could be accepted after minor revision. Please give a point-to-point response and the revised paper will be sent to the reviewer again.

·

Basic reporting

The paper is more or less an extensive technical report on some methods to compute a 2D Fourier transform of continuous functions using a kind of polar grid.

The discrete transform presented in the earlier paper is implemented and tested on a few cases.

The paper may be encouraging to others to study the weakness or strength of the proposed method.

Experimental design

The method is tested on a few cases, which have known Fourier transforms.

However, also e.g. there is a well known formula for the 2D FT of a shifted Gauss function or of a modulated one, and it should perhaps be checked if for those functions the behaviour is equally valid.

The MATLAB code (I prefer to write this program in capitals) provided appears to be quite complicated.
The matrix established by theta-matrix could be obtained much more straight (avoiding a double loop) by the simple command sequence
b2 = 2*pi/N2;
progr2= -pi + b2/2 : b2 : pi - b2/2;
theta = progr2(:)*ones(1,N1-1);
The psi-matrix is the same as the theta-matrix, so no separate code is required.
The two line code
gau = @(x) exp(- pi * ( x).^2);
f = gau(r);
should be a more elegant way of producing the sampling values, since it allows to apply the function command to a matrix of positions to get the matrix of values.

Validity of the findings

I am missing a claim: When should the new approach me preferable, faster of providing more insight. So I am taking it what it is for me: There are some arguments to try it, and it has been done, and the output is reported.

Additional comments

As a mathematician I would like to see more concrete claims or estimates. To which extent are the two matrices introduced at the beginning really (at least approximately) inverse to each other.

It should not be too difficult to compare the difference between their product and the corresponding unit matrix e.g. by doing an SVD on it, to see from where the largest deviations/errors appear. The Gauss function used for testing is certainly not in the linear span of the bad singular vectors but among the best possible ones.

---

## Round 0.3 · accepted · Accept

I recommend this paper for publication.

·

Basic reporting

This is just a revision. The authors have modified critical parts according to suggestions.

Experimental design

The MATLAB code has been adapted.

Validity of the findings

The results are certainly worth being published. The presentation is clear.